# Parity-conserving Cooper-pair transport and ideal superconducting diode in planar germanium

Marco Valentini[1] ✉, Oliver Sagi[1], Levon Baghumyan[1], Thijs de Gijsel [1,2], Jason Jung[2], Stefano Calcaterra[3], Andrea Ballabio[3], Juan Aguilera Servin [1], Kushagra Aggarwal [1,4], Marian Janik[1], Thomas Adletzberger[1], Rubén Seoane Souto [5,6], Martin Leijnse[7], Jeroen Danon[8], Constantin Schrade[9], Erik Bakkers [2], Daniel Chrastina [3], Giovanni Isella[3] & Georgios Katsaros [1] ✉

Superconductor/semiconductor hybrid devices have attracted increasing interest in the past years. Superconducting electronics aims to complement semiconductor technology, while hybrid architectures are at the forefront of new ideas such as topological superconductivity and protected qubits. In this work, we engineer the induced superconductivity in two-dimensional germanium hole gas by varying the distance between the quantum well and the aluminum. We demonstrate a hard superconducting gap and realize an electrically and flux tunable superconducting diode using a superconducting quantum interference device (SQUID). This allows to tune the current phase relation (CPR), to a regime where single Cooper pair tunneling is suppressed, creating a $\sin(2\varphi)$ CPR. Shapiro experiments complement this interpretation and the microwave drive allows to create a diode with ≈ 100% efficiency. The reported results open up the path towards integration of spin qubit devices, microwave resonators and (protected) superconducting qubits on the same silicon technology compatible platform.

III–V semiconductors have become the materials of choice for realizing high-quality hybrid devices, due to the possibility of growing epitaxial Al on top of them[1]. Gate-tunable superconducting and Andreev spin qubits[2–6], parametric amplifiers[7], highly efficient Cooper pair splitters[8–10] and a minimal Kitaev chain[11] are prominent examples of what has been achieved in the past decade. In addition, non-reciprocal devices, such as superconducting diodes have attracted a lot of interest[12], especially in Josephson junctions in the presence[13–15] or absence[16–18] of a Zeeman field and in multiterminal devices[19,20]. Diodes can be also realized in a superconducting quantum interference device (SQUID) geometry by exploiting a magnetic flux to achieve time-reversal breaking[21–23]. Such SQUIDs can be also used as a building block to create a protected superconducting qubit by engineering a $\sin(2\varphi)$ current phase relation (CPR)[24–29].

One drawback of III–V materials is their non-zero nuclear spin, which, through hyperfine interaction, drastically reduces the electron spin coherence time, limiting therefore the use of hybrid devices in combination with the spin degree of freedom[5,6]. Germanium, on the other hand, is a material which allows proximity induced superconductivity and has shown great potential for spin qubit devices[30].

[1]Institute of Science and Technology Austria, Klosterneuburg, Austria. [2]Department of Applied Physics, Eindhoven University of Technology, Eindhoven, The Netherlands. [3]L-NESS, Physics Department, Politecnico di Milano, Como, Italy. [4]Department of Materials, University of Oxford, Oxford, UK. [5]Center for Quantum Devices, Niels Bohr Institute, University of Copenhagen, Copenhagen, Denmark. [6]Instituto de Ciencia de Materiales de Madrid, Consejo Superior de Investigaciones Científicas (ICMM-CSIC), Madrid, Spain. [7]NanoLund and Solid State Physics, Lund University, Lund, Sweden. [8]Department of Physics, Norwegian University of Science and Technology, Trondheim, Norway. [9]Hearne Institute for Theoretical Physics, Department of Physics and Astronomy, Louisiana State University, Baton Rouge, USA. ✉e-mail: marco.valentini@ist.ac.at; georgios.katsaros@ist.ac.at

Induced superconductivity in germanium was first demonstrated in 0D and 1D systems[31,32]. A few years later, superconductivity was also induced in a two-dimensional Ge hole gas[33,34]. Recent works demonstrated how induced superconductivity can be improved in planar germanium, either by using a double superconducting stack[35] or by annealing platinum contacts[36]. Here, using a shallow quantum well (QW) we establish Ge/SiGe heterostructures as an alternative platform to III−V materials for hybrid devices and microwave experiments, opening therefore the path to the coexistence of semiconductor and superconducting qubits.

## Results

### Material characterization and Josephson junctions

Compressively strained Ge QWs have been deposited on relaxed, linearly graded buffers with 70% Ge content. The 18 nm thick QWs are separated from the top surface by a spacer of thickness D. The built-in in-plane compressive strain leads to charge confinement in the heavy-hole band (see Fig. 1a). Mobility ($\mu_h$) and mean free path ($l_h$) as a function of the carrier density $n_h$ are displayed in Fig. 1b for a QW with $\approx 5$ nm $Si_{0.3}Ge_{0.7}$ spacer (D5 - red) and $\approx 8$ nm (D8 - blue), respectively. At high density, D5 (D8) shows $\mu_h \approx 10,000$ cm$^2$/Vs (30,000 cm$^2$/Vs) and $l_h \approx 250$ nm (700 nm).

For creating hybrid superconductor-semiconductor devices a thin film of aluminum ($\approx 8 - 10$ nm) is deposited ex situ and at low temperature on top of the $Si_{0.3}Ge_{0.7}$ spacer (see Methods for the growth, the aluminum deposition and fabrication details). The Al has a polymorphic structure and it is not grown epitaxially on top of the $Si_{0.3}Ge_{0.7}$ spacer. Importantly, energy dispersive X-ray (EDX) data do not reveal interdiffusion of Al inside the $Si_{0.3}Ge_{0.7}$ spacer and Ge QW, see Fig. 1c.

In order to check if the superconducting properties can leak into the Ge hole gas, Josephson field effect transistors (JoFETs) were fabricated (Fig. 1d). Representative $V_{JJ}$ vs $I_{JJ}$ traces, measured in a four-terminal current-biased configuration (Fig. 1e), for D5 [red, upper plot] and D8 [blue, lower plot] are shown in Fig. 1f. The devices switch from superconducting to the dissipative regime at the gate tunable switching current $I_{sw}$. A common figure of merit used for estimating the quality of the proximity effect is the product between $I_{sw}$ and the normal state resistance $R_N$. Figure 1g reports this product as a function of the gate voltage $V_g$ for 6 different junctions with different dimensions. At high negative values of gate voltages, $I_{sw}R_N$ spans from slightly below 100 $\mu$eV to above 400 $\mu$eV depending on D and on the JoFET dimensions. These values are favorably compared to previous results obtained with Ge heterostructures hybridized by Al[33,34] and they are on par with more mature material systems[37].

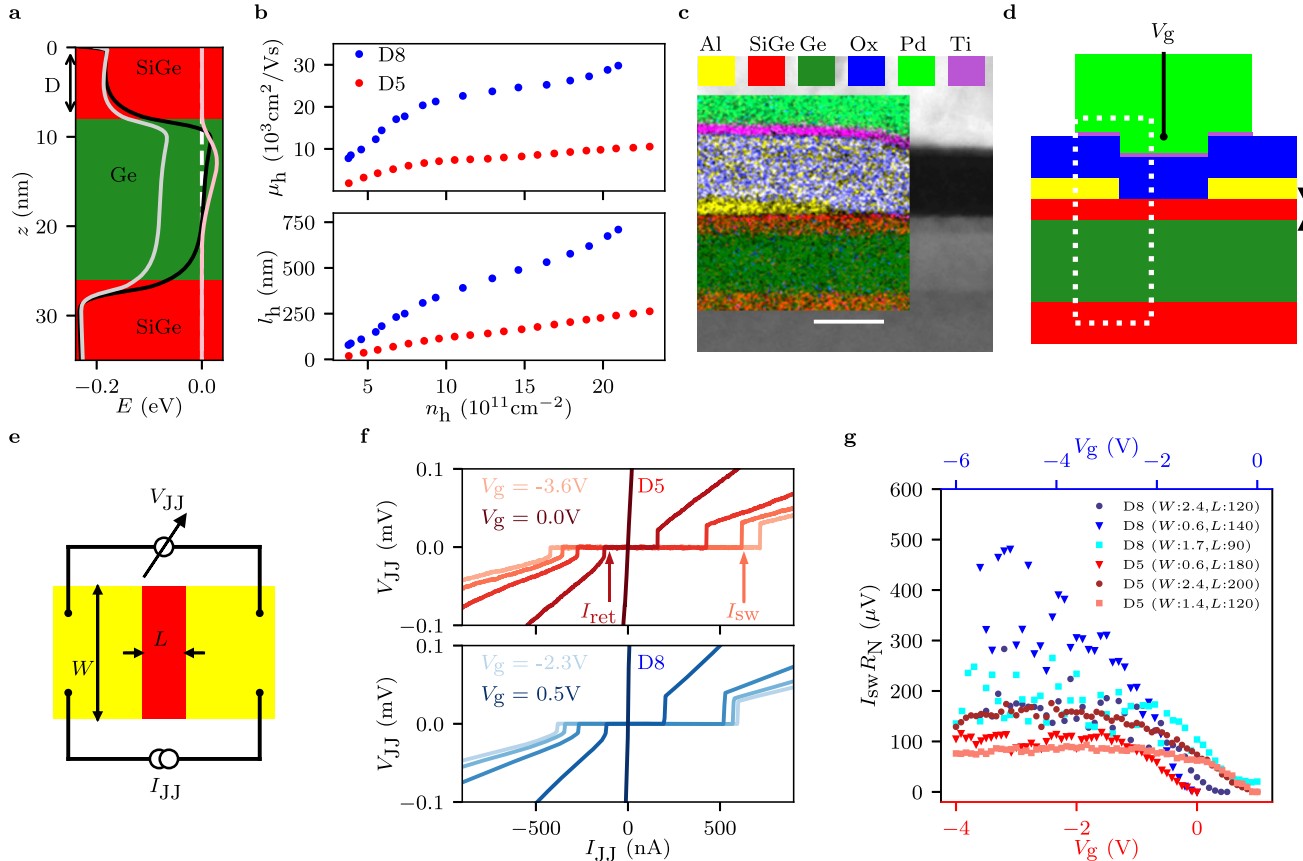

**Fig. 1 | Proximity induced superconductivity in Planar Ge. a** Heavy hole (HH) [light hole (LH)] band energy shown as a black [gray] trace along the growth direction $z$ simulated using NextNano. HHs are accumulated at the upper QW interface, as shown by the pink trace representing the HH wavefunction density plotted in arbitrary units. **b** Hole mobility $\mu_h$ [upper panel] and mean free path $l_h$ [lower panel], extracted from Hall bar measurements, as a function of carrier density $n_h$ for samples with $Si_{0.3}Ge_{0.7}$ spacer thickness (D) of 5nm (D5) and 8nm (D8). **c** TEM image of the upper part of the Ge/SiGe heterostructure. The left inset shows EDX data which confirm the absence of Al both in the spacer and in the Ge QW. The scale bar corresponds to 20 nm. **d** Cross-section sketch of a JoFET device; the dashed rectangle corresponds to the inset in **c**. The gate voltage $V_g$ is used for varying the hole carrier density in the underlying Ge QW. **e** Top-view sketch of a JoFET device with the circuit for the 4-probe measurement. The width ($W$) and the channel length ($L$) are indicated. **f** Voltage drop $V_{JJ}$ measured as a function of the applied current $I_{JJ}$ for D5 [upper panel] and for D8 [lower panel]. Lighter colors indicate lower values of $V_g$ [higher carrier density] and darker colors indicate higher values of $V_g$ [lower carrier density]. Traces are equally spaced for both panels. **g** $I_{sw}R_N$ product as a function of $V_g$ for D5 and D8 and for Josephson junctions with different dimensions as indicated in the inset. $W$ is reported in units of $\mu$m, while $L$ is in units of nm.

In sample D5, the proximity effect is expected to be more effective because the Al is closer to the Ge hole gas. It is then surprising that the measured $I_{sw}R_N$ product shown in Fig. 1g is significantly smaller for sample D5. One factor that could play a role is the fact that D8, especially at high density, is in the short ballistic regime ($L < l_h, \xi_N$), where $I_c R_N$ is expected to be equal to $\pi\Delta/e$[38]; $I_c$ is the critical current and $\xi_N = \frac{\hbar^2\sqrt{2\pi n_h}}{2m_{eff}\Delta}$ is the superconducting coherence length in the quantum well with $m_{eff}$ being the effective mass. Using $n_h = 10^{12}$cm$^{-2}$, $\Delta = 200\ \mu$eV and $m_{eff}$ to be around 10% of the electron mass[39], we estimated $\xi_N \approx 500$ nm. On the contrary, samples D5 have $L \approx l_h$, which implies a smaller $I_c R_N$[38], making it challenging to compare the $I_c R_N$ of the D8 and D5 devices directly. Moreover, the variations of $I_{sw}R_N$ in D5 and D8 as a function of the JoFET dimensions is not fully understood. For these reasons, the $I_{sw}R_N$ of such JoFET devices is not sufficient to characterize the quality of the proximity effect, especially because the switching current probability distribution is rather broad at low temperatures[40].

## Tunability of the induced superconducting gap

The above reported results demonstrate that proximity induced superconductivity can be achieved in Ge without direct contact with the superconductor, therefore avoiding metallization issues[41]. However, it is not clear to what extent the Si$_{0.3}$Ge$_{0.7}$ spacer of thickness D is influencing the value of the induced superconducting gap $\Delta^*$ and the subgap density of states. We expect that $\Delta^*$ depends on the coupling $t$ between the Al and the QW, see the sketch of Fig. 2a. Since Si$_{0.3}$Ge$_{0.7}$ acts as a tunnel barrier, $t$ should be strongly dependent on D. In other words, if D is very thin, we expect $\Delta^*$ to be similar to the parent gap of Al $\Delta$; whereas if D is very thick the two layers will be very decoupled (small $t$) and $\Delta^*$ will be quenched.

In order to investigate the dependence of $\Delta^*$ on D, tunneling spectroscopy experiments were performed to estimate the local density of states of the hybridized Ge QW, see Fig. 2b, c for the experimental layout.

The differential conductance $dI/dV$, plotted in logarithmic scale, as a function of source-drain bias $V$ and fingergates voltage $V_{dg}$ for D8 is shown in Fig. 2d. $dI/dV$ is suppressed symmetrically around $V = 0$ independently on $V_{dg}$ (Fig. 2e); this signals the presence of a superconducting gap in the hybridized Ge hole gas. Interestingly, a double peak structure is revealed. The first peak appears at $V \approx 240\mu$eV and the second at $V \approx 80\mu$eV. We interpret them as the parent gap of Al, $\Delta$, and the induced gap in the Ge hole gas $\Delta^*$. Their presence is more evident if $dI/dV$ is plotted in a linear scale, like in Fig. 2f where the line-cuts have been shifted vertically for the sake of clarity. The sub-gap conductance is suppressed by one order of magnitude compared to the above-gap value.

We now turn our attention to the $dI/dV$ of sample D5 (Fig. 2g). The difference between Fig. 2g, d is striking. The region of suppressed conductance around $V = 0$ is larger in Fig. 2g, demonstrating that the induced gap is bigger. The line-cuts, see Fig. 2h, showcase a difference between the normal-state conductance and the conductance at $V = 0$ of about two orders of magnitude, indicative of a hard gap[1]. Also for this device the double peak structure is observed (Fig. 2h, i). The parent gap appears at a similar value, namely $\Delta \approx 230\mu$eV, while $\Delta^* \approx 150\mu$eV.

## Superconducting diode effect

Having demonstrated a hard superconducting gap, we use the hybrid Al/Ge platform, to build a SQUID which acts as a gate/flux tunable superconducting diode[12] and as a generator of non-

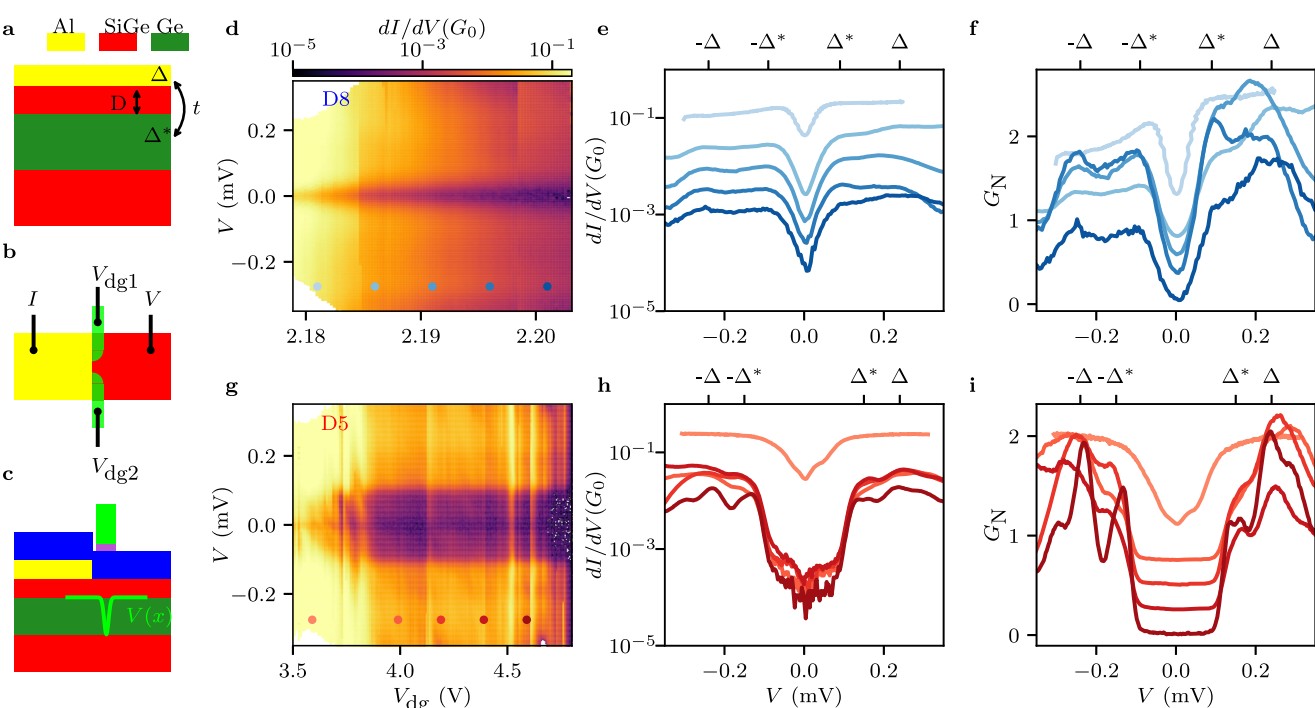

**Fig. 2 | Superconducting gap tunability. a** Sketch of the proximity effect. Al has a superconducting parent gap $\Delta$ and it is coupled to the Ge hole gas. The coupling $t$, and therefore the induced gap $\Delta^*$, depends on the thickness of the SiGe tunnel barrier, i.e., on D. **b** Top-view sketch of the device layout used to perform tunneling spectroscopy. The part of the Ge QW (right side) not covered by Al is tuned to be fully conductive and behaves like a normal metal reservoir. The two split gates are used for creating a tunnel barrier by applying voltages $V_{dg1}$ and $V_{dg2}$. The accumulation gate which covers the sample without Al on top is not depicted in the sketch. **c** Side-view sketch of **b**. The green profile is a sketch of the tunnel barrier for holes formed at the border between the conductive Ge and the hybridized Ge. **d** [**g**] $dI/dV$ as a function of $V$ and $V_{dg} = V_{dg1} + V_{dg2}$ plotted in logarithmic scale for D8 [D5]. Data for lower $V_{dg}$ are shown in Fig. S1. **e** [**h**] Line-cuts taken from **d** [**g**] at different $V_{dg}$ (see small solid circles) demonstrating a hard gap for sample D5. **f** [**i**] Line-cuts taken from **d** [**g**] plotted in a normalized scale, in which the measured $dI/dV$ is divided by the normal state conductance $G_N = (dI/dV)/G_{normal}$. The traces are shifted vertically by 0.25 $G_N$ with respect to each other.

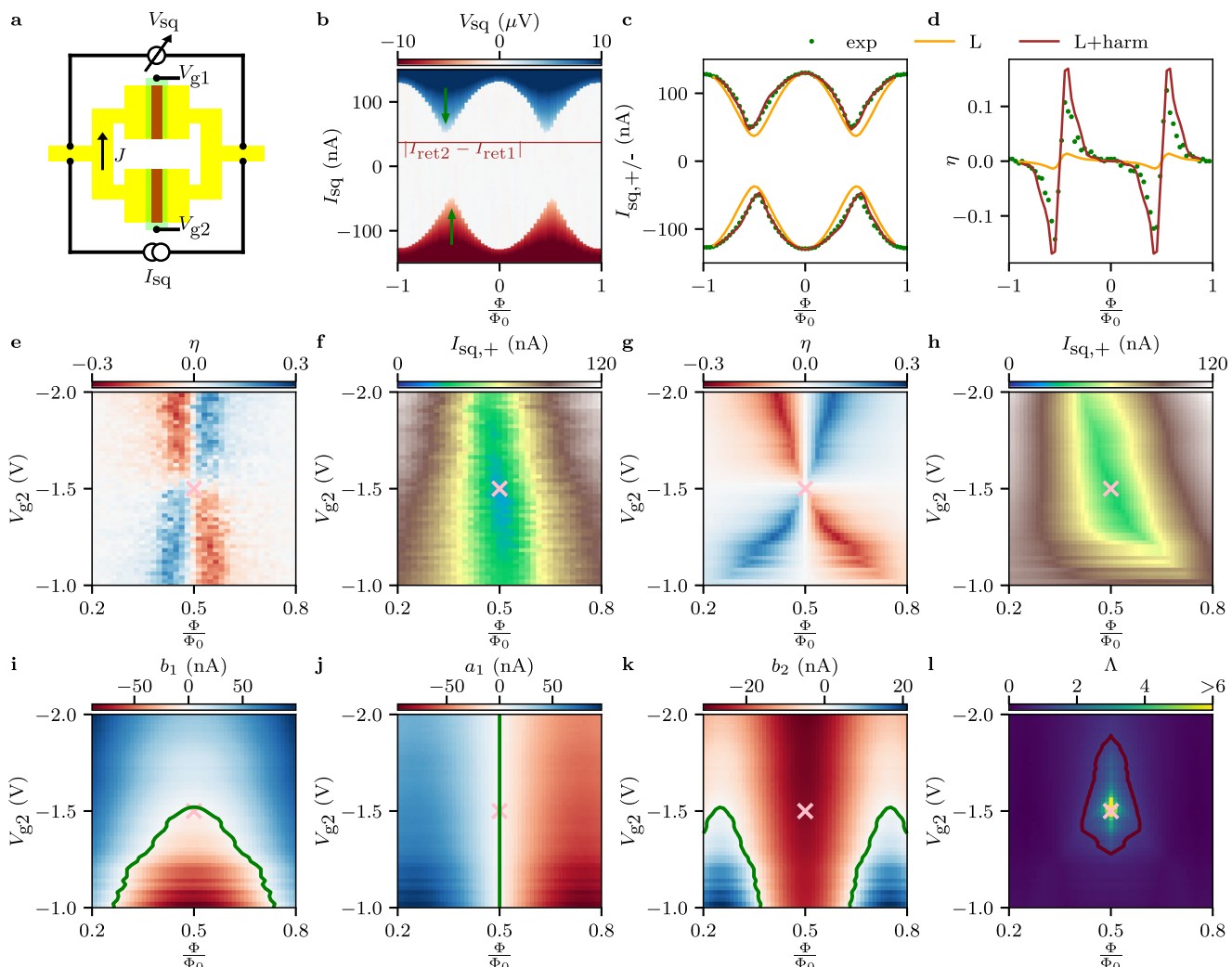

**Fig. 3 | Gate and flux tunable SDE employed as a generator of complex CPRs.**
**a** Schematic of a typical SQUID. **b** $V_{sq}$ as a function of magnetic flux $\Phi$ and $I_{sq}$. $I_{sq}$ was swept from positive values to zero and, subsequently, from negative values to zero. The SQUID lobes for positive and negative currents are asymmetric with respect to half flux and the critical current at $\frac{\Phi}{\Phi_0} = 0.5$ is larger than $|I_{ret2} - I_{ret1}|$, see the brown line. **c** Positive ($I_{sq,+}$) and negative ($I_{sq,-}$) retrapping current extracted from **b**. **d** Diode efficiency obtained from **b**. $\eta = 0$ for integer and half-integer fluxes, whereas it reaches its maxima and minima around $\frac{\Phi}{\Phi_0} = 0.5$. The orange traces represent the expected outcome if the SQUID would be composed of standard tunnel junctions and it would have a total inductance of $L = 110\,pH$. The mere addition of $L$ is not enough to match the experimental data. The red traces represent the result of the numerical fit ($K_1 = 0.658$, $K_2 = 0.122$, $K_3 = 0.102$ and higher

order terms are less than 10%), imposing $L = 110\,pH$. **e** [**f**] $\eta$ [$I_{sq,+}$] as a function of $\Phi$ and $V_{g2}$ with $V_{g1} = -1.5V$. The behavior for the switching current is shown in the Supplementary information Fig. S8. $\eta$ is always zero at $\frac{\Phi}{\Phi_0} = 0.5$, independently on $V_{g2}$, and the polarity of the diode is inverted at the balanced point (pink cross). **g** [**h**] Theoretical calculation of **e** [**f**], showing qualitatively similar behavior like the measurements. **i**–**j** First harmonic contribution extracted from **g**. The green traces highlight the points where their contribution vanishes. Just at the balanced point and at $\frac{\Phi}{\Phi_0} = 0.5$ both terms vanish. **k** Second harmonic contribution; it never vanishes at $\frac{\Phi}{\Phi_0} = 0.5$. The cosinusoidal contribution is shown in Fig. S9 along with higher order terms. **l** Ratio between second and first harmonic, $\Lambda = \frac{|b_2| + |a_2|}{|b_1| + |a_1|}$ as a function of $V_{g2}$ and $\Phi$. The red trace indicates the points where $K = 1$. In **c**, **d** and **g**–**l**, $K_1 = 0.65$, $K_2 = 0.15$, $K_3 = 0.1$ and higher order terms are less than 10%.

sinusoidal current-phase relations (CPRs). The superconducting diode effect (SDE) can appear in a simple SQUID either if its inductance $L$ is significant[42,43] or if the CPRs of the single junctions have higher order contributions, arising from Andreev bound states in a semiconducting junction[21,44] or from junctions in the dirty limit[45,46].

Figure 3a shows the schematics of a SQUID, with the underlying 4-probe current-biased electrical circuit. $I_{sq}$ is the current passing through the SQUID, $J$ is the current circulating in the SQUID and $V_{sq}$ is the measured voltage drop across the device. In the following, we always sweep $I_{sq}$ from positive/negative values to $I_{sq} = 0$, such that the retrapping current ($I_{sq,+(-)}$) is recorded for both branches. The use of the retrapping current avoids the challenges arising from the stochastic nature of the switching current[40]. Top gate voltages $V_{g1}$ and $V_{g2}$ are used to tune the retrapping current $I_{ret1}$ of JJ1 and $I_{ret2}$ of JJ2. We first

tune the device to be slightly unbalanced, i.e., $I_{ret1} = 46.5nA \neq I_{ret2} = 83.5\,nA$, see methods for understanding how $I_{ret1}$ and $I_{ret2}$ have been determined for the SQUID geometry. Figure 3b represents a SQUID measurement for such configuration, where $V_{sq}$ is recorded as a function of $I_{sq}$ and $\Phi$. $I_{sq,+(-)}$ is periodically modulated by $\Phi$, as can be clearly seen in Fig. 3c. However, two features are observed, which are not expected for a negligible inductance SQUID composed by tunnel junctions. First, the retrapping current at $\Phi = \frac{\Phi_0}{2}$ is expected to be $|I_{ret1} - I_{ret2}| = 37nA$ (see brown horizontal line in Fig. 3b), instead the measured value is around 52nA. Moreover, the SQUID pattern is not symmetric with respect to $\Phi = \frac{\Phi_0}{2}$, see green arrows in Fig. 3b. This asymmetry gives rise to a finite SDE. The diode efficiency, defined as $\eta = \frac{I_{sq,+} - |I_{sq,-}|}{I_{sq,+} + |I_{sq,-}|}$, is shown in Fig. 3d. In particular, $\eta = 0$ at integer ($\Phi = n\Phi_0$) and it changes its sign around $\Phi = \frac{n}{2}\Phi_0$. The maximum value observed for this device is around 15%.

In order to understand these results, we solve the static equation of the system:

$$\frac{I_{sq}}{2} + J = I_{JJ1}(\varphi_1),$$
$$\frac{I_{sq}}{2} - J = I_{JJ2}(\varphi_2). \tag{1}$$

$I_{JJ1}$ [$I_{JJ2}$] is the current flowing through JJ1 [JJ2] which depends on the phase difference across the junction $\varphi_1$ [$\varphi_2$]. The phase drops are related to the fluxoid quantization:

$$\varphi_2 - \varphi_1 = 2\pi \frac{\Phi}{\Phi_0} + 2\pi \frac{LJ}{\Phi_0} \tag{2}$$

For a given $\Phi$, $I_{sq,+}$ [$I_{sq,-}$] is obtained by finding the maximum [minimum] $I_{sq}$ with respect to $\varphi_1$.

First, we attempt to understand our results assuming standard sinusoidal CPRs, i.e., $I_{JJ1}(\varphi_1) = I_{ret1}\sin(\varphi_1)$ and $I_{JJ2}(\varphi_1) = I_{ret2}\sin(\varphi_2)$, and adding the inductive contribution. $L$ is composed of two terms, a geometric one $L_{geo}$ and a kinetic one $L_{kin}$; we extracted $L = 110\,pH$, see methods for details. The orange traces in Fig. 3c, d represent the theoretical prediction. It is clear that the mere addition of a realistic $L$ does not capture the full picture, especially around $\frac{\Phi}{\Phi_0} = 0.5$, where $I_{sq,+}$ and $|I_{sq,-}|$ are greatly underestimated, see Fig. S2.

Therefore, it is necessary to consider higher order harmonics for explaining our results, namely we assume that our single junction CPRs are given by:

$$I_{JJ1}(\varphi_1) = \alpha_1 I_{ret1} \sum_n (-1)^{n+1} K_n \sin(n\varphi_1)$$
$$I_{JJ2}(\varphi_2) = \alpha_2 I_{ret2} \sum_n (-1)^{n+1} K_n \sin(n\varphi_2) \tag{3}$$

where $K_n$ is the relative contribution of the n-th harmonic and we assumed that the harmonics' contribution is the same for both junctions. $\alpha_1$ [$\alpha_2$] is a dimensionless parameter which is adjusted such that $\max I_{JJ1} = I_{ret1}$ [$\max I_{JJ2} = I_{ret2}$].

The red traces in Fig. 3c, d are the outcome of a numerical fit using up to eight harmonic contributions. It is found that $K_1 = 0.66$, $K_2 = 0.12$ and $K_3 = 0.10$, while higher order terms are smaller than 10%, see Fig. S2 to understand the effect of higher order terms. We point out that also asymmetric cases would give qualitatively similar results (Fig. S3), but the amount of free parameters of the fit would increase considerably.

We now turn our attention to the gate dependence of the SDE. Our measurements show that the SDE can be tuned by the gate voltages $V_{g1}$ and $V_{g2}$. In Fig. 3e, we fix $V_{g1} = -1.5\,V$ and we study the behavior of $\eta$ while varying $\Phi$ and $V_{g2}$. When $|V_{g2}| > |V_{g1}|$, $\eta > 0$ [$\eta < 0$] for $\frac{\Phi}{\Phi_0} > 0.5$ [$\frac{\Phi}{\Phi_0} < 0.5$], while the trend is opposite if $|V_{g2}| < |V_{g1}|$. In other words, we have an inversion of the diode polarity going from one regime to the other and most importantly, the SDE completely vanishes independently of $\Phi$ when the two junctions are fully balanced ($V_{g1} \approx V_{g2}$, i.e., $I_{ret1} = I_{ret2} = I_{ret}$).

$I_{sq,+}$, plotted in Fig. 3f, does not vanish even at half flux quantum and for balanced junctions, see the pink cross in Fig. 3f; we refer to this condition as the sweet spot. This is a crucial aspect, because at the sweet spot, the first harmonic of $I_{sq}$ ($\propto \sin(\varphi)$) is completely suppressed but not the higher-order terms. We can understand this from considering Eqs. (1) at the sweet spot,

$$\frac{I_{sq}}{2} + J = \alpha K_1 I_{ret}\sin(\varphi_1) - \alpha K_2 I_{ret}\sin(2\varphi_1)$$
$$\frac{I_{sq}}{2} - J = \alpha K_1 I_{ret}\sin(\varphi_1 + \pi) - \alpha K_2 I_{ret}\sin(2\varphi_1 + 2\pi), \tag{4}$$

where for simplicity the inductance and higher order terms are neglected and $\alpha_1 = \alpha_2 = \alpha$. Therefore the CPR of the SQUID would be

$I_{sq}(\varphi_1) = -2\alpha K_2 I_{ret}\sin(2\varphi_1)$. This CPR corresponds to transport through the SQUID being governed by pairs of Cooper pairs, while the exchange of single pairs is quenched. This is the condition needed for creating a certain type protected qubit[24].

This behavior can be further understood by solving Eqs. (1), (2) and assuming to have JJs with higher order contributions. Figure 3g [h] represents the theoretical calculation of $\eta$ [$I_{sq,+}$] for a SQUID with the parameters extracted from the fit of Fig. 3c, d. The simulation results agree well on a qualitative level with the measurements. From the theoretical calculation it is possible to calculate the CPR of the SQUID and express it as a Fourier expansion:

$$I_{sq}(\varphi) \approx b_1\sin(\varphi) + b_2\sin(2\varphi) + \ldots$$
$$+ a_1\cos(\varphi) + a_2\cos(2\varphi) + \ldots \tag{5}$$

where $b_n$ and $a_n$ represents the n-th harmonic contribution and $\varphi$ is the phase drop across the SQUID. The values of the first harmonic terms $b_1$, $a_1$ and second harmonic term $b_2$ obtained from numerical simulations are shown in Fig. 3i–k.

Next, we show the theoretical prediction of the ratio between the second and first harmonic, i.e., $K = \frac{|b_2| + |a_2|}{|b_1| + |a_1|}$, see Fig. 3l. The red trace depicts the points where the first and second harmonics equally contribute to the SQUID CPR, whereas the ratio diverges close to the sweet spot.

We note that, different CPRs of the single Josephson junctions would give slightly different outcomes. However, it would not change the main conclusion that $b_1$ and $a_1$ can be completely suppressed. Moreover, the first harmonic contribution can be suppressed over a broad range of gate space, which also allows to tune the second harmonic contribution (Fig. S4).

Finally, we note that the first harmonic can be quenched by just having a high inductance and the possibility of tuning the critical currents, see Figs. S5,S6 and S7.

### Half-integer Shapiro steps and ideal SDE

The good qualitative match between the experiment and the theoretical prediction of Fig. 3 makes us confident in our interpretation of the diode data, but the CPR of the SQUID was not directly probed. The AC Josephson effect would help to further elucidate the CPR periodicity. In fact, for a standard sinusoidal CPR under microwave irradiation, the current-voltage characteristics develop voltage steps when $V = s\frac{hf_{ac}}{2e}$, the so-called Shapiro steps, where $s = 0, 1, 2, \ldots$ and $f_{ac}$ is the external applied frequency. On the contrary, if the CPR becomes $\propto \sin(2\varphi)$, signaling tunneling of pairs of Cooper pairs, steps at half-integer values also appear, i.e., $s = 0, 0.5, 1, \ldots$ [47–49]. In our case, we expect the ratio between the second and first harmonic to be maximized when the SQUID is balanced and $\Phi \approx \frac{\Phi_0}{2}$, see Fig. 3l as an example. Therefore, we would expect to observe half-integer Shapiro steps when approaching the sweet spot[21].

In order to avoid flux generated by inductive effects which might lead to similar results[50], we present results of a 30 nm-thick aluminum SQUID, which has a much smaller inductance ($L < 15\,pH$). Furthermore, a shunt resistor $R_{shunt}$ of 10-50 $\Omega$, see Fig. 4a, was added in order to create overdamped junctions, allowing therefore to measure Shapiro steps at small external frequencies, avoiding issues related to Landau-Zener transitions[51].

In the following, we study a SQUID in a balanced configuration ($I_{ret1} = I_{ret2}$) subjected to an external drive at $f_{ac} = 500\,MHz$. Figure 4b shows the differential resistance of the SQUID $dV_{sq}/dI_{sq}$ as a function of the microwave drive power $P$ and $I_{sq}$ at $\frac{\Phi}{\Phi_0} = 0.58$. If $P$ is high enough, dips corresponding to the integer Shapiro steps $s = 1, 2, 3$ appear. Similar results are obtained at $\frac{\Phi}{\Phi_0} = 0.42$, see Fig. 4c. However, the situation is different if $\frac{\Phi}{\Phi_0} = 0.5$ (Fig. 4d) a condition for which the first harmonic term should vanish. For this situation, the first half-integer

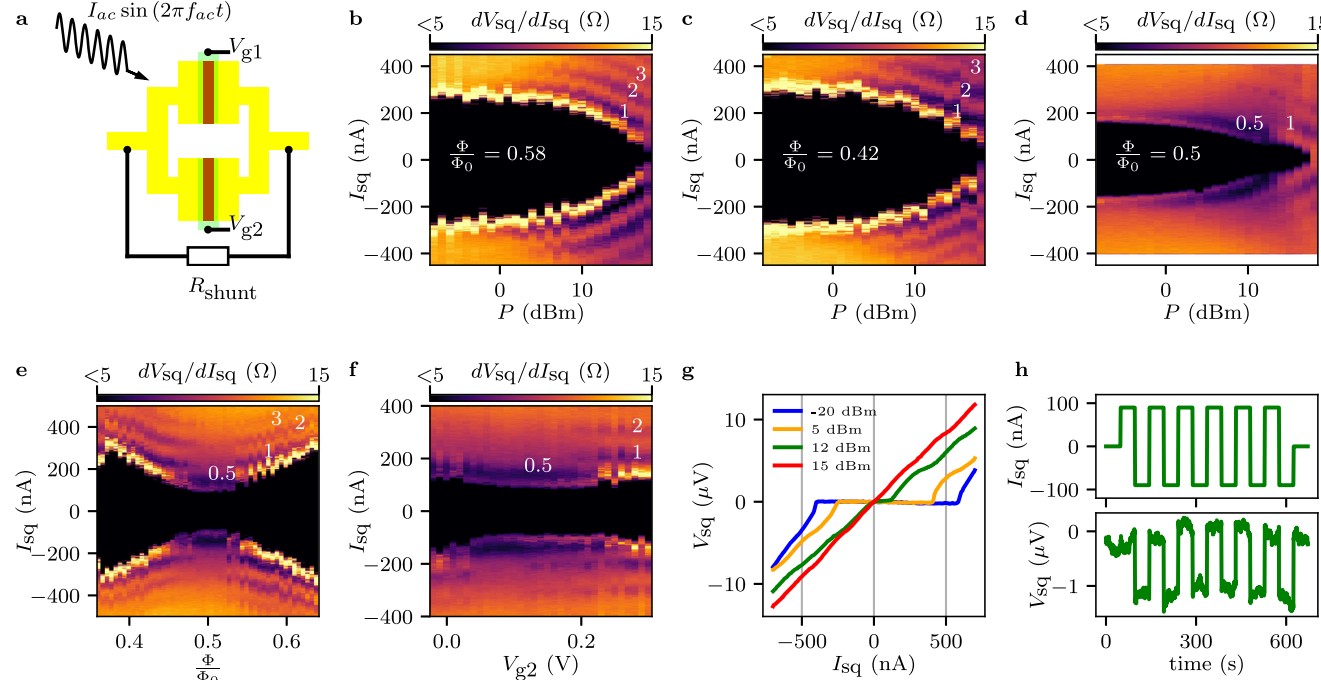

**Fig. 4 | Half-integer Shapiro steps and ideal superconducting diode.**
**a** Schematics of a typical SQUID used for Shapiro experiments with a microwave drive $I_{ac}\sin(2\pi f_{ac}t)$. **b** [**c**] Shapiro pattern for a 30nm thick room temperature deposited Al sample with $R_{shunt} = 20\Omega$ in the balanced regime ($I_{ret1} = I_{ret2} = 500$nA) with $f_{ac} = 500$MHz and at $\frac{\Phi}{\Phi_0} = 0.58$ [$\frac{\Phi}{\Phi_0} = 0.42$]. The differential resistance $dV_{sq}/dI_{sq}$ is plotted as a function of the RF power $P$ and $I_{sq}$. Dips in $dV_{sq}/dI_{sq}$ correspond to integer Shapiro steps. **d** same measurement as **b** and **c** but at $\frac{\Phi}{\Phi_0} = 0.5$. Importantly at half flux quantum, the first half-integer steps appear for low $P$. **e** Shapiro map as a function of $I_{sq}$ and $\Phi$ in the balanced regime (like for the previous plots) for $P = 9$dBm. The half-integer steps appear only close to $\frac{\Phi}{\Phi_0} = 0.5$. **f** Shapiro map as a function of $I_{sq}$ and $V_{g2}$ for $P = 9$dBm and at $\frac{\Phi}{\Phi_0} = 0.5$. The half-integer steps appear when the SQUID is close to the balanced condition, i.e., if $I_{ret2} \approx 500 \pm 40$nA. **g** and **h** show data from another 30nm thick Al sample but at $f_{ac} = 2$GHz and with $R_{shunt} = 50\Omega$ for achieving a better visibility. **g** Current-voltage characteristics for $\Phi = 0.39\Phi_0$ and for $f_{ac} = 2$GHz with $I_{ret1} = 670$nA and $I_{ret2} = 450$nA for different powers $P$. At $P = 12$dBm, $I_{sq,+}$ is 120nA, whereas $I_{sq,-} \approx 0$, i.e., $\eta \approx 1$. **h** The non-volatility of SDE at $P = 12$dBm is demonstrated by switching between the normal and superconducting behavior alternating $I_{sq}$ from 90nA to $-90$nA [upper panel]. In the lower panel, the measured voltage $V_{sq}$ is reported. A time dependent offset of $V_{sq}$, due to drift, was subtracted.

steps appears as theoretically expected (see Fig. S10 for the identification of the Shapiro steps).

This behavior is summarized in Fig. 4e where we fix $P = 9$dBm and we display $dV_{sq}/dI_{sq}$ as a function of $I_{sq}$ and $\Phi$. Far from half flux quantum, dips corresponding to integer Shapiro steps are observed; while close to $\frac{\Phi}{\Phi_0} = 0.5$ the integer steps fade and the $s = 0.5$ step becomes pronounced, see white numbers. In order to further investigate the range over which the half-integer Shapiro step is visible, we fix $\frac{\Phi}{\Phi_0} = 0.5$ and we vary $I_{ret2}$ with the gate voltage $V_{g2}$. When the SQUID is close to the balanced regime ($V_{g2} \approx 0.1$V) the first half-integer step is evident (white numbers); however it fades away if the SQUID becomes unbalanced, i.e., $V_{g2} > 0.2$V. As expected from the previous analysis, the half-integer step appears only if the device is close to the balanced position and close to half flux quantum when $I_{sq}(\varphi_1) = b_2\sin(2\varphi_1) + b_4\sin(4\varphi_1) + \ldots$. Importantly, also a second device investigated under microwave irradiation showed the same behavior, see Fig. S10.

The SDE indicates that the symmetry $I_{sq,+} = -I_{sq,-}$ is broken in our system, which also implies that the widths $\Delta I_{\pm 1}$ of the two first Shapiro steps, which eventually define $I_{sq,+}$ and $I_{sq,-}$, can be different[21]. As the position and the width of the plateaus depend on the microwave drive, one can envision tuning to a situation in which the first negative plateau would start at zero current ($I_{sq,-} = 0$) while the first positive one at a finite current ($I_{sq,+} \neq 0$). At this particular strength of the ac driving, the SQUID is expected to become an ideal SDE, i.e., $\eta \approx 1$, see theoretical analysis in ref. 52.

In order to investigate this possibility a similar SQUID, with $R_{shunt} = 50\Omega$, at $\Phi = 0.39\Phi_0$ for different drive powers $P$ was investigated (Fig. 4g). For small $P$, $I_{sq,+} > |I_{sq,-}|$ and $\eta \approx 0.18$, see blue trace.

When $P$ increases both $I_{sq,+}$ and $|I_{sq,-}|$ decrease, see orange trace. Eventually when $P$ is high enough (green trace), $I_{sq,-}$ drops to zero, whereas $I_{sq,+} \neq 0$, yielding a diode efficiency equal to 1. Moreover, we show that our device is non-volatile, namely we can switch several times from the normal-state to the superconducting branch by changing the current direction, see Fig. 4h.

## Discussion

In the past few years, planar germanium has established itself as a promising platform for spin-qubit arrays[30]. Here, we demonstrate its potential also for hybrid semiconductor-superconductor quantum devices. Inspired by more mature technologies[37], we introduced a reliable way to induce superconductivity by using shallow QWs and, to the best of our knowledge, we have realized the largest hard gap in Ge. Our method does not rely on the precise etching of the QW and/or surface treatments[35] and does not require in-situ deposition of the superconductor. Furthermore, it minimizes the Fermi velocity mismatch due to the direct contact between Ge and proximitized Ge, enhancing Andreev reflection over normal reflection.

While the shallow QWs reported in this work are of limited mobility and have a larger charge noise, which can be a challenge for the realization of scalable spin qubits, possible mitigation strategies of this problem could include a careful engineering of the semiconductor/dielectric interface[53], including the use of Ge caps[54], or growing the QWs on Ge instead of Si wafers[55]. A further solution could be to have a thin spacer in the areas where superconductivity should be induced and a thicker one in the areas where the spin qubits will be formed.

The reported large superconducting hard gap on a group IV material will enable spin qubit coupling via coherent tunneling and

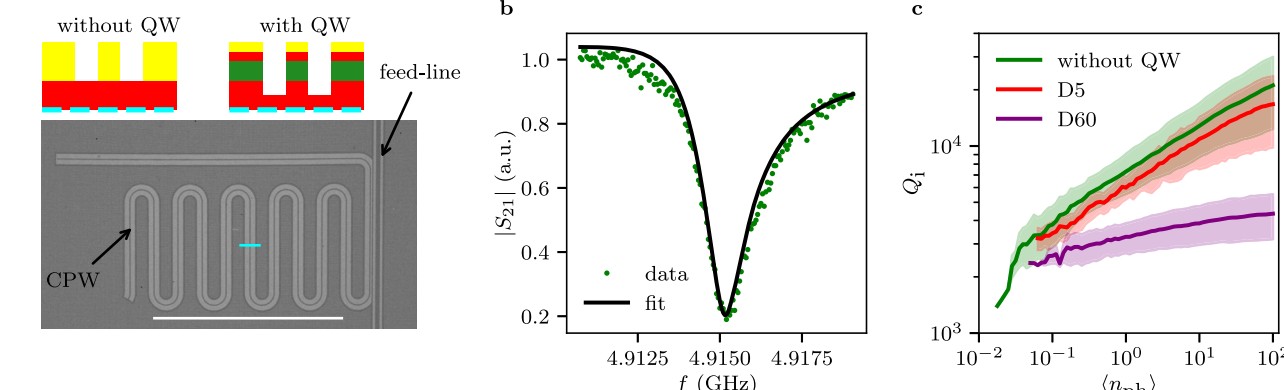

**Fig. 5 | Hybrid coplanar waveguide resonators. a** Scanning electron microscopy image of a typical notch-type CPW resonator. The top insets show sketches of the CPW cross-section, at the position of the cyan trace, when the QW has been etched away (left) and when the resonator is fabricated on top of the QW (right). In the latter case, the feedline, the resonator and the ground plane are formed on top of the Ge QW. The scale bar corresponds to 500 $\mu m$. **b** $|S_{21}|$ signal as a function of the frequency $f$ around the resonance frequency of a device without QW. The black line is an algebraic fit[58]. **c** Internal quality factor $Q_i$, averaged for multiple 4GHz < $f_r$ < 5.5GHz resonators (Table S1), as a function of the average photon number $\langle n_{ph} \rangle$ for samples without QW, D5 and D60. The shaded areas correspond to the standard deviation of the data. We notice that the proximitized Ge QW in the samples D5 leads to small additional losses, compared to the without QW resonator samples. However, if the Ge QW gets decoupled from the Al (D60), we observe a lower $Q_i$, suggesting that $Q_i$ is limited by losses in Ge.

cotunneling processes that involve (crossed) Andreev reflection[56,57]. In addition, the realized gate and flux-tunable superconducting diode can suppress the first harmonic term, making it therefore an interesting building block for creating protected superconducting qubits with semiconductor materials[24–28]. In order to realize such qubits, superconducting resonators are key elements. A $\lambda/4$ notch-type resonator is shown in (Fig. 5a) (see methods for details). The upper-left inset of Fig. 5a depicts the cross-section of the resonator, pointing out that the Ge QW has been completely etched away prior to the resonator fabrication. Figure 5b shows the transmission amplitude $|S_{21}|$ as a function of the probe frequency $f$. The internal quality factors $Q_i$ were extracted[58] and found to be around 7000 [20000] for $\langle n_{ph} \rangle \approx 1$ [100]; demonstrating the microwave compatibility of the used Ge/SiGe heterostructures. Interestingly, just slightly smaller $Q_i$ values were extracted also for superconducting resonators fabricated on Ge/SiGe heterostructures where the Ge QW has been removed just in the gap between the central conductor and the ground plane, showing that the proximitized Ge does not lead to significant losses. The above demonstrated microwave compatibility of the used Ge/SiGe heterostructures opens a path towards spin-photon experiments[59], gate tunable transmon qubits[2–4] and superconducting spin qubits in group IV materials[6] and allow us to envision the transfer of quantum information between different types of qubits, all realizable on planar Ge.

After submission of our manuscript we became aware of similar works dealing with the superconducting diode effect in interferometer devices[60–62].

## Methods

### Growth and Al deposition

Strained Ge QW structures were grown by low-energy plasma-enhanced chemical vapor deposition on forward-graded buffers[63] with $Si_{0.3}Ge_{0.7}$ caps of 5 and 8 nm above the 18 nm Ge QW. These nominal QW and cap thicknesses vary across the wafer due to the intensity profile of the focused plasma. Thicknesses were verified by comparing high-resolution x-ray diffraction $\omega - 2\theta$ scans with dynamical simulations based on a smoothed QW profile[63]. This same composition profile was constructed within the `NextNano` 1-d Poisson-Schrödinger solver, along with a dielectric layer and top Schottky contact, in order to generate the band profile and wavefunction density shown in Fig. 1a. The Ge/SiGe heterostructures are cut in pieces of 6x6 mm². A 3 min buffered oxide etch (7:1 ratio) removes the native surface oxide of the

diced samples, after which they are transferred to a molecular beam epitaxy chamber. The samples are cooled down to 110K by active liquid-nitrogen cooling. Subsequently, Al is deposited at a growth rate of 5.5 Å/min. Immediately after growth, samples are transferred in-situ to a chamber equipped with an ultrahigh-purity $O_2$ source where they are exposed to $10^{-4}$mbar of $O_2$ for 15 min. The formed oxide layer prevents subsequent retraction of the metal film as the sample warms up to room temperature under ultrahigh vacuum conditions.

### Sample fabrication

**10 nm-thick, cold deposited aluminum samples.** A mesa of around 60nm depth is obtained by first removing Al with Transene D and then by etching the heterostructure with a $SF_6$-$O_2$-$CHF_3$ reactive ion etching process. In a second step, Al is selectively etched away using Transene D in order to create the Josephson junction or tunneling spectroscopy devices. Then, for tunneling spectroscopy devices, normal metal ohmic contacts are created by argon milling the SiGe spacer followed by a deposition of 60 nm platinum at an angle of 5°. Finally, 9–18 nm plasma assisted aluminum oxide is deposited on top of all the sample at 150 °C and then Ti/Pd gates are evaporated. For some devices, two layers of top-gates were needed.

**30 nm-thick, room-temperature deposited aluminum samples.** A mesa of around 60nm depth is obtained by etching the heterostructure with a $SF_6$-$O_2$-$CHF_3$ reactive ion etching process. The sample is then submerged for 15s in buffered HF and, subsequently, the 30nm Al film is deposited. Gates are patterned like for the 10nm-thick sample. Importantly, this technique allows to fabricate devices without the need of wet etching for removing the superconductor.

**CPW resonator, without QW.** The QW is removed by a $SF_6$-$O_2$-$CHF_3$ reactive ion etching process. Subsequently, the CPW resonator, the feed-line and the ground plane are written by electron beam lithography followed by a 25 nm-thick Al deposition at room temperature.

**CPW resonator, with QW.** Electron beam lithography is performed on a sample with low temperature deposited aluminum. The area between the ground plane and the signal line is exposed and after development the Al is removed by transene D etching. Finally, and before removing the resist the Ge QW is etched away by a $SF_6$-$O_2$-$CHF_3$ reactive ion etching process.

**Table 1 | Estimation of SQUID inductance**

|                          | 10 nm Al | 30 nm Al |
|--------------------------|----------|----------|
| $R_\square$              | 12.2Ω    | 0.75Ω    |
| $T_c$                    | 1.9K     | 1.4K     |
| $L_{kin,\square}$        | 8.9pH    | 0.75pH   |
| $L_{kin} + L_{geo}$      | 110pH    | 13pH     |

$R_\square$ and $T_c$ were estimated from 4-probe current biased measurements. The SQUIDs consist of approximately 12 squares.

### Inductance estimation

The total inductance $L$ of the measured SQUIDs has two contributions: the geometric one ($L_{geo}$) and the kinetic one ($L_{kin}$). For the $L_{geo}$ we approximated the device to a loop with a radius $R$ and with the wire diameter $d$

$$L_{geo} = \mu_0 R \left[ \ln\left(\frac{16R}{d}\right) - 2 \right] \qquad (6)$$

where $\mu_0$ is the magnetic vacuum permeability and we assumed the relative magnetic permeability $\mu_r = 1$. Typical values of our SQUID geometry are $R \approx 1.25\,\mu m$ and $d \approx 0.7\,\mu m$, which gives $L_{geo} \approx 2pH$. However, in order not to underestimate this contribution, $L_{geo}$ is assumed to be as large as $\approx 5pH$. As regards the kinetic inductance per square $L_{kin,\square}$, it was estimated from the values of the superconducting gap and the square normal-state resistance $R_\square$[64]:

$$L_{kin,\square} = \frac{h}{2\pi^2} \frac{R_\square}{\Delta} \qquad (7)$$

where $\Delta$ is estimated from the critical temperature $T_c$, i.e., $\Delta = 1.76 k_B T_c$ with $k_B$ being the Boltzmann constant. The results are summarized in Table 1.

### Estimation of the retrapping currents in the SQUID geometry

We note that the sum of the retrapping currents measured in isolation ($I_{ret1,iso}$ and $I_{ret2,iso}$), i.e., with the other junction pinched off, in the absence of a shunt resistor is always smaller than the retrapping current of the squid at $\Phi = 0$. This difference is attributed to the fact that the SQUID has a smaller resistance, which leads to lower dissipation[65] and, as a result, a higher retrapping current. Therefore, we assume an even redistribution of retrapping currents such that $\frac{I_{ret1,iso}}{I_{ret2,iso}} = \frac{I_{ret1}}{I_{ret2}}$ and $I_{ret1} + I_{ret2} = I_{sq,+}(\Phi = 0)$. This approach was used to estimate the retrapping currents for Fig. 3.

### Data availability

All experimental data included in this work are available at https://zenodo.org/records/10119346.

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

## Acknowledgements

We acknowledge Alexander Brinkmann, Alessandro Crippa, Francesco Giazotto, Andrew Higginbotham, Andrea Iorio, Giordano Scappucci, Christian Schonenberger, and Lukas Splitthoff for helpful discussions. We thank Marcel Verheijen for the support in the TEM analysis. This research and related results were made possible with the support of the NOMIS Foundation. It was supported by the Scientific Service Units of ISTA through resources provided by the MIBA Machine Shop and the nano-fabrication facility, the European Union's Horizon 2020 research and innovation programme under Grant Agreement No 862046, the HORIZON-RIA 101069515 project, the European Innovation Council Pathfinder grant no. 101115315 (QuKiT), and the FWF Projects #P-32235, #P-36507 and #F-8606. For the purpose of open access, the authors have applied a CC BY public copyright licence to any Author Accepted Manuscript version arising from this submission. R.S.S. acknowledges Spanish CM "Talento Program" Project No. 2022-T1/IND-24070. J.J. acknowledges European Research Council TOCINA 834290.

## Author contributions

M.V. and T.G. fabricated the transport devices and performed the measurements under the supervision of G.K. M.V. did the data analysis under the supervision of G.K. L.B. and O.S. fabricated and measured the CPW. M.J. and O.S. developed the microwave technology for the Ge/SiGe heterostructures. K.A. contributed to the transport measurements and the device fabrication. J.A.S. fabricated the Hall bars. T. A. made the Shapiro measurements possible. M.V. performed the simulations with input from C.S. and R.S.S. S.C., A.B., D.C., and G.I. were responsible for the Ge QW growth, Hall bar measurements, and NextNano simulations. J.J. and E.B. were responsible for the growth of low-temperature Al and TEM data. M.L. and J.D. contributed to the interpretation of the experimental results. M.V. and G.K. wrote the manuscript with input from all the coauthors.

## Competing interests

The authors declare no competing interests.
