## [Peer Review File · Nature Communications]

Parity-conserving Cooper-pair transport and ideal superconducting diode in planar Germanium.REVIEWER COMMENTS

Reviewer #1 (Remarks to the Author):

The manuscript by Valentini et al. reports about Josephson junctions obtained in Ge hole gas proximitized by Al. The authors characterize the induced gap and junctions parameters, and study the superconducting diode effect both with DC transport and with radio-frequency excitation. I found the first part of the paper -the one about characterization of Ge hole gas-based junctions- very interesting owing to the crucial role that such innovative material might play in quantum computing and to their advantages compared to III-V materials. However, the second part of the paper, focusing on the SC diode effect is not very convincing, as discussed below. Given the importance that the authors themselves attribute to the diode effect (as one deduces from the title) I would like that the authors clarify the points below before making a final assessment about the opportunity of a publication in Nature Communications.

1- It is not clear to me how exactly the authors determine I_{ret1} and I_{ret2} , independently? How do the authors can set $I_{ret1}=60\text{nA}$ and $I_{ret2}=69\text{nA}$ (and be absolutely sure about these values)? From the text it seems that the authors do determine the two I_{ret} independently of the I_{sq} pattern, in fact they write that they expect a retrapping current of $69-60=9\text{nA}$ at half quantum of flux, and instead they have much more.

A possibility, if I might guess, is that the authors first pinch off the first junction, then they record the gate voltage 2 that produces a certain I_{ret2} . Then, they pinch off junction 2 and determine the gate voltage 1 that produces a certain I_{ret1} . Then they set the two found gate voltages. Is this the case? If yes, this method would require perfect reproducibility in the gate operation (no hysteresis of the gates).

This point is very important because balancing a SQUID is a nontrivial task, as discussed in the similar work [arXiv:2306.05467 Ciaccia et al.] (see final comment below).

2-Why do the authors consider I_{ret} (IV swept from finite bias to zero) and not I_{crit} (from zero to finite bias)? Taking I_{ret} heating effects might play an important role, determining an important increase of the electron temperature. This unusual choice must be explained. Also, papers in the literature have discussed the important difference between switching and retrapping crit current in Josephson diodes (see e.g. Steiner et al., PRL 130 177002 (2023), which might be cited).

3-It is known that the skewness of the CPR (i.e. its higher harmonics content) is hidden by screening effects, which mimic very well a tilted CPR. The authors claim that they could really disentangle the two effects. I'm not very convinced by their analysis. Let us take Fig3c. The authors claim to have determined once for all the right L being 110pH , then they determined the right I_{ret1} and I_{ret2} (see questions above). With this, they claim that the simple L cannot match the data, which is the argument to invoke the higher harmonics. This is a strong statement, for which strong evidence would be needed. In particular:

3a the inductance might have been underestimated.... The authors should report which L would produce the best fit of data in Fig.3c without higher harmonics. If, and only if, this value is unacceptably high, then the authors could start considering other options (higher harmonics).

3b the I_{ret1} 2 could be not exactly determined: the modelling in Fig 3c critically depends on the precise knowledge of the two I_{ret} values.

3c. If higher harmonics are really present and do play a role, the best way to demonstrate this would be to intentionally suppress the higher harmonics with higher temperature. If the effect is due to screening, the kin inductance will increase with T, while the higher harmonics of the junction CPRs are suppressed with T. This is a measurement that I would highly recommend.

3d. A possible way to understand whether higher harmonics come from the CPR of individual junctions or from trivial SQUID screening-related effect is to measure Shapiro steps of each individual junction (alternatively pinching off the other one by gating), as it was done in [arXiv:2306.05467 Ciaccia et al.]: higher harmonics should produce half integer Shapiro steps.

-4 The authors admit that some L is present, and I agree (this is also visible from the misalignment of the green arrows in Fig3b). I think that this invalidates the argument used to

state, starting from Eqs 4, that I_{sq} is just proportional to the second harmonics. with a LJ screening term [Eq 2] added in the first harm sine of the second equation, the cancellation of the 1st harm term does not take place. Moreover, the LJ term added in the argument depends again on the two supercurrents, which makes the phase dependence complicated, nonsinusoidal, with the appearance of higher harmonics.

5- it seems to me that the authors themselves are not convinced by their own arguments about the presence of higher harmonics when they write "Even if the SDE observed in this work would not be given by a CPR with higher order harmonics but by a high inductance value, the main message would not be altered."

I do not think that the message will not be altered. The effect of the inductance would be trivial and well known.

"In fact, also in this case, due to the gate tunability of the semiconductor JJ, it is possible to find a sweet spot where ... $I_{sq}(\phi_1) = b_2 \sin(2\phi_1) + b_4 \sin(4\phi_1)$ ".

Could the authors demonstrate this statement? If I add a LJ term in the argument of the first sine of the second Eq 4, then it seems to me that there is no way to fully cancel the first harmonics term by just tuning I_{ret} (even if one could tune both separately). I would like the authors to explain this point.

6- This work appeared as preprint nearly simultaneously with [arXiv:2306.05467 Ciaccia et al.], which studies a different material but it makes use of similar experimental methods to study the same effect. Given the large overlap of the two works and the almost simultaneous appearance, I would emphasize the existence of a similar study of 4e-charge supercurrent on a different material by properly citing it (perhaps with a note added at the end).

A minor thing:

The same symbol K is used both for the Fourier coefficient of the CPR and as ratio first to second harmonic.

Reviewer #2 (Remarks to the Author):

The authors of Radio frequency driven superconducting diode and parity conserving Cooper pair transport in a two-dimensional germanium hole gas focus on superconductor semiconductor hybrid devices. In particular, the authors focus on germanium, which is a material that gained relevance in the development of spin qubit systems.

The authors claim "The reported results open up the path towards monolithic integration of spin qubit devices, microwave resonators and (protected) superconducting qubits on a silicon technology compatible platform."

Unfortunately, I am not convinced this work defines the advancement needed to recommend for publication and particularly I am not convinced the data is supporting this conclusion.

First of all, the authors seem to follow prior work by using aluminum as the superconductor. Their novelty is reducing the SiGe barrier thickness, to reduce the distance between the quantum well and the superconductor. However, whereas in this work the focus is on reducing the thickness, the trend for spin qubit devices is opposite. The reason to use deeper quantum wells is a reduced charge noise and improved uniformity. Indeed the measured mobilities in the devices under study are inferior compared to deeper quantum wells used for spin qubit devices. This shortcoming is mentioned at the end of the article, but the proposed solutions do not seem obvious and simple changes that will suddenly improve these aspects. In other words, I would certainly recommend for publication if the authors could demonstrate the same results on a high-mobility low-charge noise shallow quantum well.

Unfortunately, I am also not convinced by the resonator development. For example, the internal q-factors seem to be rather low (e.g. how do they compare to works in silicon where spin-photon coupling has already been demonstrated?).

If the authors can address these comments I would be willing to review this work again.

Some minor comments:

Please quantify the discussion on a hard superconducting gap. How hard is the reported gap, how does it compare to other work, and in particular how hard should it be for the anticipated experiments. For example, is the observed gap hard enough for a high-fidelity gate based on a Cooper pair splitter? It may be challenging to address this, but then it is also important to be clear that further improvements may be needed for these types of experiments.

The paragraph on page 3 starting with 'In sample D5' is confusing. It seems to end stating that these devices are not sufficient to study proximity effects. If so, then why starting the discussion in the first place. I also recommend to substitute proximity effect with IcRN if that is what is meant. Please clarify this paragraph.

Page 3, sentence: "These values are ... mature material systems."
Please also compare to the work on annealing platinum contacts.

Please change the sentence in the discussion: "The reported large ... coupling via superconductors." A large superconducting gap is not necessary for spin-qubit coupling via superconductors. For instance, this objective has already been demonstrated using a capacitive interaction. A large gap may indeed be needed for Cooper pair splitters.

Reviewer #3 (Remarks to the Author):

Referee report for Valentini et al.,

The authors investigate induced superconductivity in Ge 2DEGs. They find a "hard induced gap", and they also demonstrate that the system can be used with RF technology (superconducting resonators on the substrate). Their most important finding is the observation of SC diode effect in a SQUID configuration. They attribute this to the non-sinusoidal current-phase relation, and also investigate the dependence on the gate voltage (critical current of the junctions). However, probably what I find the nicest is the correlation with the Shapiro measurements. Altogether I find the manuscript interesting and a very thorough and comprehensive analysis, altogether a very nice and important work. The methodology and data analysis is sound. I have some doubts though, which need to be answered before I can suggest publication in Nature Communications.

- It has been recently shown by Tosato et al. (Comm. Materials, 2023., 4) that a hard superconducting gap can be engineered in germanium 2DEG. I would be interested in the novelty of this work compared to that paper.

- One such novelty could be the diode effect in a SQUID. However a very similar finding appeared on arxiv by the Basel group (see Ciaccia et al., arXiv:2306.05467 and arXiv:2304.00484), where they correlated to 2ϕ CPR with radiation experiments.

Altogether I would say the authors should clarify in their response and the manuscript as well, what is the novelty in their work. Still, I find this work very important.

- I do not fully understand the title of the work: "Radio frequency driven superconducting diode". The diode works fully in DC as well. Does the title refer to the Shapiro measurements?

- The authors discuss the non-sinusoidality of the CPR, but if I did not miss it, I did not see any CPR measurements. Was there any CPR measurements performed (the SQUID would allow this).

- How did the authors get the mobility? Is it a field effect mobility or was it obtained in Hall-bars?
- I did not fully get the discussion on the $I_C R_N$ product. As a remark, it would be nice to collect all the numbers (junction length, mean free path) to the same part of discussion. If I got it well, both junctions are ballistic or close to and short. I do not expect a large difference between them. If it is diffusive, they could compare with the diffusive mean free path and the $I_C R_N$ product with the Thouless energy.
- I also do not get the two gap argument. If the region below the SC is proximitized the SC parent gap should not be visible, since the barrier is towards the proximity gap. It should be visible, if there is a barrier inbetween the SC and the proximity region. Or is there a current path, which goes directly from the SC electrode to the barrier (not going through the proximity region)?
- If I understand well, in Eq. 4 second order harmonics were substituted for the junctions CPR. What is equation 5 then? Is it a Fourier expansion of the SQUID?
- I did not understand why the retrapping and not the switching currents are used. Could the authors explain it?

Reply to Referee report for manuscript "Radio frequency driven superconducting diode and parity conserving Cooper pair transport in a two-dimensional germanium hole gas."

September 17, 2023

1 Reply to referee 1

The manuscript by Valentini et al. reports about Josephson junctions obtained in Ge hole gas proximitized by Al. The authors characterize the induced gap and junctions parameters, and study the superconducting diode effect both with DC transport and with radio-frequency excitation. I found the first part of the paper -the one about characterization of Ge hole gas-based junctions- very interesting owing to the crucial role that such innovative material might play in quantum computing and to their advantages compared to III-V materials. However, the second part of the paper, focusing on the SC diode effect is not very convincing, as discussed below. Given the importance that the authors themselves attribute to the diode effect (as one deduces from the title) I would like that the authors clarify the points below before making a final assessment about the opportunity of a publication in Nature Communications.

We thank the referee for the positive feedback and for the careful reading of our manuscript. Below, we reply to all the points raised in the report.

1. It is not clear to me how exactly the authors determine I_{ret1} and I_{ret2} , independently? How do the authors can set $I_{ret1}=60\text{nA}$ and $I_{ret2}=69\text{nA}$ (and be absolutely sure about these values)? From the text it seems that the authors do determine the two I_{ret} independently of the I_{sq} pattern, in fact they write that they expect a retrapping current of $69-60=9\text{nA}$ at half quantum of flux, and instead they have much more. A possibility, if I might guess, is that the authors first pinch off the first junction, then they record the gate voltage 2 that produces a certain I_{ret2} . Then, they pinch off junction 2 and determine the gate voltage 1 that produces a certain I_{ret1} . Then they set the two found gate voltages. Is this the case? If yes, this method would require perfect reproducibility in the gate operation (no hysteresis of the gates). This point is very important because balancing a SQUID is a nontrivial task, as discussed in the similar work [arXiv:2306.05467 Ciaccia et al.] (see final comment below).

The guess of the referee about the determination of I_{ret1} and I_{ret2} is correct. We first determined I_{ret1} as a function of V_{g1} by pinching off the second Josephson junction (JJ). Then we pinched off the first JJ and determined I_{ret2} as a function of V_{g2} . However, balancing the SQUID devices is a straightforward task as described below.

In Fig. R1 we show similar experiments like the one shown in Fig.3e, performed at two different times and after having swept V_{g1} considerably. Fig. R1b shows η as a function of V_{g2} and the perpendicular field B with $V_{g1} = -1.5\text{V}$. As expected from the pinch off in Fig. R1a, the diode effect vanishes and, consequently the SQUID is balanced when $V_{g2} \approx V_{g1} \approx -1.5\text{V}$. Fig. R1c shows the same measurement as **b** but after having swept V_{g1} from -2V to -0.8V and then back to -1.8V . The balanced point is changed by around 0.05V in the gate space, which corresponds to around 3 nA difference (less than 10%). Such uncertainty does not change the conclusion of our work, as also discussed later, and it does not prevent us to balance the SQUID. In fact, while performing the experiment described in Fig. R1b, we fix V_{g1} and we vary V_{g2} . When η vanishes independently on the value of B , we know that at that point $I_{ret1} = I_{ret2}$, and we stop sweeping V_{g2} . The SQUID is then balanced and stable as demonstrated by the measurements in Figs.4b, **c** and **d** of the manuscript, where Φ was swept. Such measurements took around 31 hours and the device remained in the balanced configuration, demonstrating its stability and the fact that balancing the SQUID is an easy task for devices realized in shallow Ge/SiGe heterostructures.

We have added a paragraph in the methods "Estimation of the retrapping currents in the SQUID geometry" to further explain how the retrapping currents were extracted and we decided to show configuration of Fig. R1b

Figure R1: **Gate reproducibility.** **a** $I_{\text{ret}1}$ and $I_{\text{ret}2}$ as a function of V_{g1} and V_{g2} , respectively. **b** η as a function of V_{g2} and magnetic field B . **c** same measurement as **b** but taken at a different time and with a higher resolutions in V_{g2} and B .

Figure R2: SQUID pattern obtained by recording the switching current instead of the retrapping current.

in the main text as a function of V_{g2} instead of $I_{\text{ret}2}$.

2. Why do the authors consider I_{ret} (IV swept from finite bias to zero) and not I_{crit} (from zero to finite bias)? Taking I_{ret} heating effects might play an important role, determining an important increase of the electron temperature. This unusual choice must be explained. Also, papers in the literature have discussed the important difference between switching and retrapping crit current in Josephson diodes (see e.g. Steiner et al., PRL 130 177002 (2023), which might be cited).

We decided to use the retrapping current instead of the switching current because the switching current has a stochastic behaviour, as explained for instance in reference [1]. As an example, in Fig. R2 we show a typical SQUID experiment for a thin-film Aluminum sample recording the switching current instead of the retrapping current. The stochastic behaviour is evident and it is impossible to extract reliably the diode efficiency. One would need to record the value of the switching current many times and extract the mode of the switching current distribution, which would make all the measurements longer and less reliable. Notably, the stochastic behaviour was much less pronounced for a sample with thicker Aluminum contacts and therefore lower kinetic inductance and the diode experiment was conducted by recording both the retrapping and the switching current, as shown in Fig.ED8a-d. Importantly, the same values of diode efficiency are obtained by recording the retrapping and the switching currents. In particular, in both cases, it is possible to reach η of approximately 0.3.

We note that the observed hysteretic behaviour cannot be explained by the standard RCSJ model. We believe it is caused by heating effects, as pointed out by the reviewer, and carefully examined in ref. [2]. Following the reviewer's suggestion we have added a citation to Steiner et al in our manuscript and we added the following sentence explaining why we recorded the retrapping current: "The use of the retrapping current avoids the challenges arising from the stochastic nature of the switching current [1]."

3a. It is known that the skewness of the CPR (i.e. its higher harmonics content) is hidden by screening effects, which mimic very well a tilted CPR. The authors claim that they could really disentangle the two effects. I'm not very convinced by their analysis. Let us take Fig3c. The authors claim to have determined once for all the right L being 110pH, then they determined the right $I_{\text{ret}1}$ and $I_{\text{ret}2}$ (see questions above).

Figure R3: **Fit with higher-order components for different SQUID configurations and cases with $L = 110\text{pH}$.** The first and second rows correspond to two different gate voltage configurations. The results of the fits are reported in Table 1.

With this, they claim that the simple L cannot match the data, which is the argument to invoke the higher harmonics. This is a strong statement, for which strong evidence would be needed. The inductance might have been underestimated... The authors should report which L would produce the best fit of data in Fig.3c without higher harmonics. If, and only if, this value is unacceptably high, then the authors could start considering other options (higher harmonics).

To the best of our knowledge, we have not underestimated the inductance. We extracted the kinetic inductance by using two different methods and they both gave the same result. First, as mentioned in the methods, we estimated the kinetic inductance per square by measuring the normal-state resistance of a four-probe sample. Second, we estimate the inductance value by measuring the resonance frequency f_r of the coplanar waveguide resonator described in Fig. 5. The resonator of length l is modeled as a distributed LC circuit and its f_r has two contributions f_{geo} and f_{kin} (see for instance ref [3]), i.e.

$$\frac{1}{f_r^2} = \frac{1}{f_{\text{geo}}^2} + \frac{1}{f_{\text{kin}}^2} \quad (1)$$

where $f_{\text{geo}} = (2l\sqrt{L_{\text{geo},\square}C})^{\frac{1}{2}}$ and $f_{\text{kin}} = (2l\sqrt{L_{\text{kin},\square}C})^{\frac{1}{2}}$. $L_{\text{geo},\square}$ and C are determined solely by the geometry and they are calculated by standard Sonnet simulations. Therefore, by measuring f_r we estimated that $L_{\text{kin},\square} \approx 10\text{pH}$, as extracted also from the first method.

We note that similar values of kinetic inductance per square have been reported also for a planar InAs heterostructure with 7nm of Al [F. Nichele et al., Phys. Rev. Lett. 124, 226801 (2020)]. In that work they report a total kinetic inductance of 290pH for 48 squares.

As strong evidence that the mere presence of inductance cannot explain our results, in Fig. ED2a we would need an inductance of around 5 nH to match the experimental data. Such value is 50 times higher than the value measured with the two methods described above, and, thus, it is unacceptably high.

3b. the $I_{\text{ret}1}$ 2 could be not exactly determined: the modelling in Fig 3c critically depends on the precise knowledge of the two I_{ret} values.

	K_1	K_2	K_3	K_4	K_5	K_6	K_7	K_8	$I_{\text{ret}1}$ (nA)	$I_{\text{ret}2}$ (nA)
Fig. R3a	0.620	0.1644	0.1069	0.057	0.031	0.0129	0.0073	$< 1e - 5$	66.8	58.1
Fig. R3b	0.6533	0.12161	0.090	0.040	0.0399	0.01667	0.0024	0.00819	75	50
Fig. R3c	0.7523	0.0631	0.0799	0.01391	0.0499	0.01388	0.0274	$1e - 4$	85	40
Fig. R3d	0.658	0.122	0.102	0.036	0.042	0.013	0.018	0.009	83.5	46.5
Fig. R3e	0.700	0.090	0.095	0.030	0.033	0.025	0.019	0.010	90	40
Fig. R3f	0.572	0.165	0.110	0.069	0.042	0.027	0.012	0.005	72	58

Table 1: Fit results from Fig. R3. Importantly, the presence of the higher harmonic terms does not significantly change when the retrapping currents are varied, supporting once more the conclusion of our work that higher harmonic terms are present.

Figure R4: **Temperature dependence for a thick Aluminum SQUID device.** **a** and **b** η as a function of Φ taken at different temperature T . For both configurations, η decreases with T .

We agree that the modelling of Fig. 3c depends on the values $I_{\text{ret}1}$ and $I_{\text{ret}2}$. Deviations from their values will yield different harmonic components but it does not change our conclusion that higher harmonic terms are necessary in order to explain our experimental findings. In Fig. R3 we report two SQUID configurations, configuration 1 in the upper row and configuration 2 in the lower row. For each configuration, we assumed that $L = 110$ pH and we analyzed three different cases of $I_{\text{ret}1}$ and $I_{\text{ret}2}$. Importantly, the fit yields a qualitatively similar result, see Table 1. Now, Fig.3 b-c-d of the main text correspond to configuration 2, which has been thoroughly analyzed for different retrapping configurations.

3c. If higher harmonics are really present and do play a role, the best way to demonstrate this would be to intentionally suppress the higher harmonics with higher temperature. If the effect is due to screening, the kin inductance will increase with T, while the higher harmonics of the junction CPRs are suppressed with T. This is a measurement that I would highly recommend.

Following the referee's advice, we performed the diode experiment for a thick Aluminum sample (lower inductance $L \approx 15$ pH) both at $T = 50$ mK and at $T = 400$ mK. Fig. R4 shows the η as a function of Φ for two different gate voltages configuration. While for the configuration shown on the right side, with higher retrapping currents, the diode efficiency has just slightly dropped, for the configuration shown in the left side, with higher values of retrapping current, η is significantly smaller at higher temperature. These measurements therefore suggest that the diode behaviour observed in our work strongly depends on the presence of higher harmonic terms.

3d. A possible way to understand whether higher harmonics come from the CPR of individual junctions or from trivial SQUID screening-related effect is to measure Shapiro steps of each individual junction (alternatively pinching off the other one by gating), as it was done in [arXiv:2306.05467 Ciaccia et al.]: higher harmonics should produce half integer Shapiro steps.

We performed Shapiro experiments on single junction devices and we, sometimes, observed half-integer steps as shown in Fig. R5. However, we decided not to use them as a proof of higher-order harmonics contributions, because there are other phenomena, such as non-equilibrium effects, which could also generate

Figure R5: Half-integer Shapiro steps for a single Josephson junction device.

half-integer steps [4]. Therefore, we decided to employ Rf radiation only to probe the CPR in the SQUID configuration. In fact, for the SQUID we knew exactly that they should arise at the sweet spot ($I_{\text{ret}1} \approx I_{\text{ret}2}$ and $\Phi = \frac{\Phi_0}{2}$).

4. The authors admit that some L is present, and I agree (this is also visible from the misalignment of the green arrows in Fig3b). I think that this invalidates the argument used to state, starting from Eqs 4, that I_{sq} is just proportional to the second harmonics. with a LJ screening term [Eq 2] added in the first harm sine of the second equation, the cancellation of the 1st harm term does not take place. Moreover, the LJ term added in the argument depends again on the two supercurrents, which makes the phase dependence complicated, nonsinusoidal, with the appearance of higher harmonics.

Based on our replies to the previous questions, we hope that we have convinced the referee that our inductance is small and it cannot explain our experimental data. Furthermore, also the presence of higher-order harmonics creates the misalignment of the green arrows in Fig. 3b, see Fig. ED2c and Fig. 3c of reference [5].

As we have shown in Figs. ED5 and ED6 of the original manuscript, the first term of the SQUID CPR (b_1) can also be suppressed for a finite inductance and without higher harmonics at half flux quantum. This can be also seen from the blue traces in Fig. R6. However, if the inductance would be high, like Figs. R6d and h, one would need to properly tune the device. Namely, b_1 , in Figs. R6d and h, would vanish for a particular value of $I_{\text{ret}2}$, with $I_{\text{ret}2} > I_{\text{ret}1}$. Thus, it is also possible to cancel the first harmonic in the CPR of the SQUID for finite inductance.

5. it seems to me that the authors themselves are not convinced by their own arguments about the presence of higher harmonics when they write “Even if the SDE observed in this work would not be given by a CPR with higher-order harmonics but by a high inductance value, the main message would not be altered.” I do not think that the message will not be altered. The effect of the inductance would be trivial and well known. “In fact, also in this case, due to the gate tunability of the semiconductor JJ, it is possible to find a sweet spot where ... $I_s q(\phi_{i1}) = b_2 \sin(2\phi_{i1}) + b_4 \sin(4\phi_{i1})$ ”. Could the authors demonstrate this statement? If I add a LJ term in the argument of the first sine of the second Eq 4, then it seems to me that there is no way to fully cancel the first harmonics term by just tuning I_{ret} (even if one could tune both separately). I would like the authors to explain this point.

We admit that the phrasing “the main message would not be altered” was not the proper choice. We wanted to point out that the superconducting diode effect in a SQUID and the suppression of the first harmonic can be either obtained because of the presence of higher harmonics or even in their absence in the case of a significant inductance, which however is not the case in our work. We have changed the phrasing of the sentence to: “We note that, different CPRs of the single Josephson junctions would give slightly different outcomes. However, it would not change the main conclusion that b_1 and a_1 can be completely suppressed, see Figs. ED9. Moreover, the first harmonic contribution can be suppressed over a broad range of gate space, which also allows to tune the second harmonic contribution (Fig. ED7). Finally, we note that the first harmonic can be quenched by just having a high inductance and the possibility of tuning the critical currents. However, the presence of a not negligible inductance might hinder the possibility of achieving parity conserving transport, see Fig.ED9.”

As shown in Figs. ED3, ED4, ED5 and ED6 of the original manuscript, it is possible to cancel both a_1 and b_1 , even in the presence of a large inductance. Due to gate tunability, it is always possible to tune the devices such that $b_1 = 0$, see blue traces of Fig. R6 even in the absence of higher-order terms. However, if the inductance is not negligible, see for instance Fig. R6h, b_3 indeed does not vanish at $I_{\text{ret}2}$ values at which

Figure R6: **Calculation of the first three harmonics of the SQUID CPR at half flux quantum by varying $I_{\text{ret}2}$ and with $I_{\text{ret}1} = 100$ nA.** In the upper row, higher-order terms are neglected ($K_2 = K_3 = 0$). In the lower row, higher-order terms are considered ($K_1 = 0.7, K_2 = 0.2, K_3 = 0.1$). In the first column it is assumed $L = 0$, in the second $L = 10$ pH (as for the thick-Aluminum samples), in the third $L = 100$ pH (as for the thin-Aluminum samples) and in the fourth row $L = 1$ nH.

$b_1 = 0$. Therefore, the SQUID CPR will be $I_{\text{sq}}(\psi_1) = b_2 \sin(2\psi_1) + b_3 \sin(3\psi_1) + b_4 \sin(4\psi_1)$. On the contrary, if $L = 0$ and we have higher-order terms (see Fig. R6e), all the odd terms vanish at the same value of $I_{\text{ret}2}$ and we have parity conserving transport $I_{\text{sq}}(\psi_1) = b_2 \sin(2\psi_1) + b_4 \sin(4\psi_1)$. Fig. R6f and g depict the case for $L = 10$ pH (30 nm thick Al samples) and for $L = 100$ pH (10 nm thick Al samples), respectively. For these inductance values, b_1 and b_3 vanish almost at the same value.

6. This work appeared as preprint nearly simultaneously with [arXiv:2306.05467 Ciaccia et al.], which studies a different material but it makes use of similar experimental methods to study the same effect. Given the large overlap of the two works and the almost simultaneous appearance, I would emphasize the existence of a similar study of 4e-charge supercurrent on a different material by properly citing it (perhaps with a note added at the end).

Following the suggestion of the reviewer we have added the following note at the end of our work. "After submission of our manuscript we became aware of similar works dealing with the superconducting diode effect in interferometer devices." There we cite the second works of Ciaccia et al., and two more works (Refs. 57 and 58), which have recently appeared in the literature, [Li et al, arXiv:2306.08478 (2023), Matsuo et al., Nature Physics (2023)]. Whereas the previous Ciaccia et al. work was already cited. Moreover, we also added the following references [15-18] as they are relevant for our work.

A minor thing: The same symbol K is used both for the Fourier coefficient of the CPR and as ratio first to second harmonic.

We used Λ to represent the ratio between the second and first harmonics instead of K .

2 Reply to referee 2

The authors of Radio frequency driven superconducting diode and parity conserving Cooper pair transport in a two-dimensional germanium hole gas focus on superconductor semiconductor hybrid devices. In particular, the authors focus on germanium, which is a material that gained relevance in the development of spin qubit systems. The authors claim "The reported results open up the path towards monolithic integration of spin qubit devices, microwave resonators and (protected) superconducting qubits on a silicon technology compatible platform." Unfortunately, I am not convinced this work defines the advancement needed to recommend for publication and particularly I am not convinced the data is supporting this conclusion.

We thank the reviewer for their critical comments which have helped us to clarify the significance of our work and further improve its quality. In the current work we have investigated the potential of shallow Ge/SiGe quantum wells for hybrid semiconductor-superconductor devices. We have realized the largest hard gap reported for Germanium, shown the microwave compatibility of Ge/SiGe heterostructures for cQED related experiments, realized a superconducting diode, with an efficiency which can be tuned by means of a microwave drive to 100% and demonstrated the tunneling of pairs of Cooper pairs.

We would like to further point out that the Ge/SiGe heterostructures used in this work have a peak mobility which is similar (or higher) to that of silicon metal-oxide-semiconductor devices which are one of the leading platforms for spin qubit experiments [6]. However, in contrast to the Silicon devices, here we can induce also superconductivity via the proximity effect. Other platforms that allow this are group III-V materials. Due to the hyperfine interaction the dephasing times for spin qubits in III-V materials is in the 10 ns range [7] and also in state of the art III-V material heterostructures the mobilities are very similar to those reported in our work [8]. Based on the above facts, we believe that we can indeed state that our results open up the path towards co-integration of spin qubits and hybrid devices.

First of all, the authors seem to follow prior work by using aluminum as the superconductor.

Indeed, the novelty of our results does not lie in the choice of the superconductor. By using the most common superconductor we have measured the largest, to the best of our knowledge, induced hard gap in Germanium.

Their novelty is reducing the SiGe barrier thickness, to reduce the distance between the quantum well and the superconductor. However, whereas in this work the focus is on reducing the thickness, the trend for spin qubit devices is opposite. The reason to use deeper quantum wells is a reduced charge noise and improved uniformity. Indeed the measured mobilities in the devices under study are inferior compared to deeper quantum wells used for spin qubit devices. This shortcoming is mentioned at the end of the article, but the proposed solutions do not seem obvious and simple changes that will suddenly improve these aspects. In other words, I would certainly recommend for publication if the authors could demonstrate the same results on a high-mobility low-charge noise shallow quantum well.

We agree with the reviewer that the trend for spin qubits in terms of spacer thickness is opposite to what we have used in our current work. The goal of the present work is a thorough investigation of the proximity induced superconductivity in planar Germanium and the realization of high quality hybrid devices. On the other hand, we would like to point out, that for spin qubit devices in planar Ge it is not proven that deeper quantum wells indeed improve the dephasing times. In fact, the Veldhorst group in Delft has realized spin qubits in planar Ge for a QW buried 22nm and dephasing times of up to 833 ns were reported at a field of 500mT [9]. This value is larger than what was reported by the same group for 55 nm deep Ge quantum wells; the measured dephasing times were less than 500ns at 500mT (<https://repository.tudelft.nl/islandora/object/uuid:97c4ea24-9672-4e0b-b7a5-e3a48258c871?collection=research>). We further point out that for Ge hut wires with just a 2nm spacer, dephasing times approaching 200ns have been reported [10]. Therefore for spin qubit devices the interface between the heterostructure and the oxide seems to be the most critical factor affecting their coherence. However, we agree that in order to improve the uniformity one should work with deeper quantum wells showing a lower percolation density, i.e. Ge based Ge/SiGe heterostructures, but this would be a completely different work and is clearly out of the scope of the present paper. Eventually, the solution might be to have a thin spacer in the areas where superconductivity should be induced and a thicker spacer in the areas where the spin qubits will be formed. However, this is will be the focus of future research efforts.

Unfortunately, I am also not convinced by the resonator development. For example, the internal q-factors seem to be rather low (e.g. how do they compare to works in silicon where spin-photon coupling has already been demonstrated?). If the authors can address these comments I would be willing to review this work again.

We agree that the reported quality factors are lower than what has reported for resonators on highly resistive silicon or sapphire. However, they are large enough to allow cQED experiments. In the hole spin-

photon coupling work in Silicon, which reported the largest coupling constant, the internal quality factor was 530 [11].

Please quantify the discussion on a hard superconducting gap. How hard is the reported gap, how does it compare to other work, and in particular how hard should it be for the anticipated experiments. For example, is the observed gap hard enough for a high-fidelity gate based on a Cooper pair splitter? It may be challenging to address this, but then it is also important to be clear that further improvements may be needed for these types of experiments.

Our usage of the term "hard gap" was based on the work of Chang et al. [12], in which they defined it to mean that the subgap conductance is at least two orders of magnitude lower than the above-gap conductance. In our experiment, we observe at least such a difference in the conductances (our subgap conductance is in fact within the noise level of the used lockin amplifier, about $10^{-4} G_0$), which is similar to state-of-the-art III-V hybrid devices [13] and Ge/SiGe heterostructures made using the annealing technique [14].

One quality measure that was recently estimated in an experiment using a Cooper pair splitter with a similar gap hardness was a spin correlation of 96 percent [15]. In general, however, it is indeed challenging to predict very precisely the effect of the hardness of the gap on two-qubit gate fidelities, especially given the crucial role played by the dimensionality and cleanness of the superconducting coupler (see, e.g., the discussion in Ref. [PRL 111, 060501 (2013)]). Assuming, for simplicity, the best-case scenario of a low-dimensional ballistic superconductor, a very rough estimate of the qubit-qubit coupling provided by crossed Andreev reflection yields $\gamma_{\text{CAR}} \sim t^2 \rho_{\text{N}}$, where t is the tunnel coupling energy between the qubit levels and the superconductor and ρ_{N} the superconductor's normal density of states, see [PRL 111, 060501 (2013)] (we also assume the superconducting coherence length to be long enough and the qubit levels to be tuned close to the center of the gap). A finite subgap density of states will lead to leakage in the form of single-particle tunneling between the qubit levels and the superconductor, with a rate of the order $\gamma_{\text{leak}} \sim t^2 \rho_{\text{sg}}$, with ρ_{sg} the subgap density of states. Altogether, in this best-case scenario one can thus expect an infidelity that scales roughly as $\sim G_{\text{sg}}/G_{\text{N}}$, i.e., as the ratio of the subgap and normal conductances of the superconductor.

The paragraph on page 3 starting with 'In sample D5' is confusing. It seems to end stating that these devices are not sufficient to study proximity effects. If so, then why starting the discussion in the first place. I also recommend to substitute proximity effect with IcRN if that is what is meant. Please clarify this paragraph.

Following the remark of the reviewer we have slightly reformulated this paragraph. The point we wanted to make is that a large IcRN product does not imply that a superconducting gap with a low subgap conductance has been induced in the Ge hole gas, as one could possibly assume.

Page 3, sentence: "These values are ... mature material systems." Please also compare to the work on annealing platinum contacts.

Compared to the work in which platinum has been annealed, our measurements show a larger superconducting gap. We have added this information to the manuscript.

Please change the sentence in the discussion: "The reported large ... coupling via superconductors." A large superconducting gap is not necessary for spin-qubit coupling via superconductors. For instance, this objective has already been demonstrated using a capacitive interaction. A large gap may indeed be needed for Cooper pair splitters.

We agree that the absence of subgap conductance is more important. However, it is still relevant to have a large gap because when qubit operations are performed one might want to detune the quantum dots and then one always has to stay inside the gap. With a small gap one has a very limited bias window. Furthermore, a small gap might mean a lot of thermally excited quasiparticles which is also detrimental for qubit operations.

3 Reply to referee 3

The authors investigate induced superconductivity in Ge 2DEGs. They find a “hard induced gap”, and they also demonstrate that the system can be used with RF technology (superconducting resonators on the substrate). Their most important finding is the observation of SC diode effect in a SQUID configuration. They attribute this to the non-sinusoidal current-phase relation, and also investigate the dependence on the gate voltage (critical current of the junctions). However, probably what I find the nicest is the correlation with the Shapiro measurements. Altogether I find the manuscript interesting and a very thorough and comprehensive analysis, altogether a very nice and important work. The methodology and data analysis is sound. I have some doubts though, which need to be answered before I can suggest publication in Nature Communications.

We thank the referee for the positive evaluation of our work and the questions which allowed us to further emphasize the novelty of our work.

It has been recently shown by Tosato et al. (Comm. Materials, 2023., 4) that a hard superconducting gap can be engineered in germanium 2DEG. I would be interested in the novelty of this work compared to that paper.

In the work by Tosato et al. work, superconducting contacts were created by annealing Pt into SiGe. This is a very interesting and novel approach, however it requires an annealing step and the superconducting properties depend on the annealing temperature, the process followed and the amount of Pt deposited, i.e. on several tuning knobs. Our approach is very straightforward and allows to tune the value of the superconducting gap by simply changing the thickness of the spacer. In other words, we have developed a reliable and easy-to-realize recipe to induce hard gap superconductivity in a Germanium quantum well. Moreover, in Tosato et al., a superconducting gap of around $70 \mu\text{eV}$ was realized, whereas in our case it is around $150 \mu\text{eV}$.

The main difference, besides the value of the superconducting gap, between the two approaches is, that the annealing technique leads to lateral contacts with the Ge QW, i.e. the QW beneath the superconductor has transformed into PtSiGe. Our method minimizes the Fermi velocity mismatch as the Ge channel is in direct contact with proximitized Ge, enhancing further Andreev over normal reflection. Furthermore, with our lift-off method it is easy to define precisely small structures. This is more difficult with the annealing technique as the Pt does not diffuse just vertically. Such precision is very important for creating hybrid quantum dot structures of small dimensions, which could be used for the realization of Andreev spin qubits and Kitaev chains. Finally, by adjusting the coupling between the superconductor and the QW below, i.e. the SiGe spacer thickness, the properties of the proximitized semiconductor can be tuned [16].

One such novelty could be the diode effect in a SQUID. However a very similar finding appeared on arxiv by the Basel group (see Ciaccia et al., arXiv:2306.05467 and arXiv:2304.00484), where they correlated to 2ϕ CPR with radiation experiments. Altogether I would say the authors should clarify in their response and the manuscript as well, what is the novelty in their work. Still, I find this work very important.

We are aware of the two works mentioned, and we have included citations for them at the end of our manuscript. Here, we report all the novelties of our work:

- First demonstration of hard gap in Ge without annealing and/or surface treatment [17].
- Largest reported hard gap in Ge.
- First demonstration of proximity effect tunability by varying the SiGe spacer thickness.
- First demonstration of microwave compatibility of Germanium quantum well heterostructures.
- Demonstration of the superconducting diode effect in a hybrid SQUID geometry, similar to arXiv:2304.00484.
- Demonstration of the possibility of realizing parity-conserving transport, simultaneously with arXiv:2306.05467.
- First demonstration of an ideal superconducting diode ($\eta = 100\%$) induced by the application of a microwave drive. Theoretical works will follow to investigate this novel approach to boost the diode efficiency.

Following the suggestion of the reviewer, we have emphasized the novelties in our revised manuscript.

I do not fully understand the title of the work: “Radio frequency driven superconducting diode”. The diode works fully in DC as well. Does the title refer to the Shapiro measurements?

We acknowledge that we had difficulties to find a proper title able to cover all the novelties of our work. As explained above, we showed that by applying a microwave drive we can tune the diode efficiency, for this reason we decided to call “Radio frequency **driven** superconducting diode ...”. However, after reading carefully the

comments of the reviews we have realized that they consider different aspects of our work as the most novel elements. Therefore, we have adapted the title of the manuscript to cover most of the aspects discussed in our work.

The new title of our manuscript reads: Proximity effect, ideal superconducting diode and parity-conserving Cooper-pair transport in microwave-compatible Ge/SiGe heterostructures.

The authors discuss the non-sinusoidality of the CPR, but if I did not miss it, I did not see any CPR measurements. Was there any CPR measurements performed (the SQUID would allow this).

We are aware that DC unbalanced SQUIDS are often used to extract the CPR, however the extraction of the CPR might give inaccurate results. In fact, the accuracy of the CPR measurement is given by the asymmetry in the derivatives of the CPRs instead of the critical currents, as properly explained in the recent publication Babich et al. (acs.nanolett.3c01970).

In order to avoid such complications, we assumed that the two junctions have a similar CPR and we directly probe the sum of the two CPRs, like in Fig. 3b.

How did the authors get the mobility? Is it a field effect mobility or was it obtained in Hall-bars?

The mobility is measured via Hall-bars. We added a sentence in the figure caption.

did not fully get the discussion on the $I_C R_N$ product. As a remark, it would be nice to collect all the numbers (junction length, mean free path) to the same part of discussion. If I got it well, both junctions are ballistic or close to and short. I do not expect a large difference between them. If it is diffusive, they could compare with the diffusive mean free path and the $I_C R_N$ product with the Thouless energy.

As explained in our reply to the second reviewer, the point we wanted to make is that a large $I_C R_N$ product does not imply that a superconducting gap with a low subgap conductance has been induced in the Ge hole gas, as one could possibly assume. We have reformulated this paragraph. Moreover, information about the Josephson junctions geometry are summarized in the inset of Fig. 1g.

I also do not get the two gap argument. If the region below the SC is proximitized the SC parent gap should not be visible, since the barrier is towards the proximity gap. It should be visible, if there is a barrier inbetween the SC and the proximity region. Or is there a current path, which goes directly from the SC electrode to the barrier (not going through the proximity region)?

Indeed there is a barrier (SiGe cap) between the proximitized quantum well and the superconductor. If we apply a bias bigger than the induced superconducting gap ($V > \Delta^*$), then one more type of tunnel process directly into the superconductor is allowed. This process can take place either directly through the tunnel barrier or first via the proximitized region and then via the barrier. Importantly, the density of states in the superconductor is much larger than in the proximitized semiconductor, so it makes sense that the current increases even if the tunnel amplitude is small.

If I understand well, in Eq. 4 second order harmonics were substituted for the junctions CPR. What is equation 5 then? Is it a Fourier expansion of the SQUID?

Yes, equation 5 is the Fourier expansion of the SQUID.

I did not understand why the retrapping and not the switching currents are used. Could the authors explain it?

The reason why we used the retrapping current instead of the switching currents is that the switching current has a stochastic behaviour and, therefore, is impossible to extract reliably without recording for every configuration the switching current distribution. We point out that we also measured a thick-film aluminum sample where the switching current had a less stochastic behavior, and in this case the diode efficiency extracted from the retrapping and switching current were similar, see Fig. ED8a-d. More information can be found in the reply to referee 1.

References

- [1] Haxell, D. Z. *et al.* Measurements of phase dynamics in planar josephson junctions and squids. *Physical Review Letters* **130**, 087002 (2023).
- [2] Courtois, H., Meschke, M., Peltonen, J. & Pekola, J. P. Origin of hysteresis in a proximity josephson junction. *Physical review letters* **101**, 067002 (2008).
- [3] Phan, D. *et al.* Detecting induced $p \pm i p$ pairing at the al-inas interface with a quantum microwave circuit. *Physical Review Letters* **128**, 107701 (2022).
- [4] Iorio, A. *et al.* Half-integer shapiro steps in highly transmissive insb nanoflag josephson junctions. *arXiv preprint arXiv:2303.05951* (2023).
- [5] Souto, R. S., Leijnse, M. & Schrade, C. Josephson diode effect in supercurrent interferometers. *Physical Review Letters* **129**, 267702 (2022).
- [6] Scappucci, G., Taylor, P. J., Williams, J. R., Ginley, T. & Law, S. Crystalline materials for quantum computing: Semiconductor heterostructures and topological insulators exemplars. *MRS Bulletin* **46**, 596–606 (2021). URL <https://doi.org/10.1557/s43577-021-00147-8>.
- [7] Nadj-Perge, S., Frolov, S. M., Bakkers, E. P. A. M. & Kouwenhoven, L. P. Spin-orbit qubit in a semiconductor nanowire. *Nature* **468**, 1084–1087 (2010). URL <https://doi.org/10.1038/nature09682>.
- [8] Mayer, W. *et al.* Superconducting proximity effect in epitaxial al-inas heterostructures. *Applied Physics Letters* **114**, 103104 (2019).
- [9] Hendrickx, N. W., Franke, D. P., Sammak, A., Scappucci, G. & Veldhorst, M. Fast two-qubit logic with holes in germanium. *Nature* **577**, 487–491 (2020). URL <https://doi.org/10.1038/s41586-019-1919-3>.
- [10] Watzinger, H. *et al.* A germanium hole spin qubit. *Nature communications* **9**, 3902 (2018).
- [11] Yu, C. X. *et al.* Strong coupling between a photon and a hole spin in silicon. *Nature Nanotechnology* **18**, 741–746 (2023). URL <https://doi.org/10.1038/s41565-023-01332-3>.
- [12] Chang, W. *et al.* Hard gap in epitaxial semiconductor–superconductor nanowires. *Nature Nanotechnology* **10**, 232–236 (2015). URL <https://doi.org/10.1038/nnano.2014.306>.
- [13] Valentini, M. *et al.* Nontopological zero-bias peaks in full-shell nanowires induced by flux-tunable andreev states. *Science* **373**, 82–88 (2021).
- [14] Tosato, A. *et al.* Hard superconducting gap in germanium. *Communications Materials* **4**, 23 (2023).
- [15] Wang, Q. *et al.* Triplet cooper pair splitting in a two-dimensional electron gas. *arXiv preprint arXiv:2211.05763* (2022).
- [16] Adelsberger, C., Legg, H. F., Loss, D. & Klionvaja, J. Microscopic analysis of proximity-induced superconductivity and metallization effects in superconductor-germanium hole nanowires. *arXiv preprint arXiv:2306.06944* (2023).
- [17] Aggarwal, K. *et al.* Enhancement of proximity-induced superconductivity in a planar ge hole gas. *Physical Review Research* **3**, L022005 (2021).

REVIEWER COMMENTS

Reviewer #1 (Remarks to the Author):

I find the authors' responses to the Referees' requests satisfactory, and the changes to the text appropriate. I therefore recommend publication in Nature Communications.

Reviewer #2 (Remarks to the Author):

The authors have made effort to address all comments, but one critical point remains to be addressed.

As the authors state, the novelty of the work is not in the materials development, but in the variation of the SiGe barrier thickness. The authors acknowledge that this reduction is a strong compromise in the uniformity of the quantum well, which is visible in for example a reduced mobility and percolation density. The sentence that the interface is crucial in determining the coherence seems to be at odds with the preceding section about dephasing times. However, I do agree with the authors that charge noise is most likely coming from the interface. Indeed the reduced uniformity and the reduced charge noise is the reason for the spin qubit community to focus on deeper quantum wells.

I therefore remain of the option that the gain in the superconducting proximity effect is not sufficient enough compared to the reduction in uniformity to recommend this work for publication. Moreover, in my opinion this particular concern is not clearly addressed and a critical section highlighting all compromises and challenges is missing.

Some minor points:

In the outlook the sentence 'The reported large superconducting hard gap on a group IV material will pave the way towards spin qubit coupling via superconductor' is not clear. A capacitive interaction can be sufficient for coupling to resonators. The authors mean hybrid systems based on tunnel coupling between superconductors and semiconductors.

Reviewer #3 (Remarks to the Author):

Referee report for Valentini et al.,

The authors did several changes in the manuscript, however, in certain places I found them a bit reluctant to do further analysis, measurements. However, if they can answer my minor comments I suggest the paper to be published.

- Regarding the larger SC gap and the versatility of the method I understand and accept the reasoning of the authors.
- Regarding the question of CPR measurements that authors write, that it is technically challenging (since the asymmetry that has to be used depends on the transmission), however it is possible. They gave similar arguments to the Shapiro measurements. The temperature dependence however, is a good indication. Though I think these measurements could have been done, I think all other findings point to having higher harmonics, therefore I accept their answer.
- In principle, the retrapping is also a stochastic process – one can measure retrapping current distribution. Do I understand well, that the authors claim, that the retrapping was reproducing well

in their measurements?

- I have suggested in the case of the $I_c R_n$ product discussion to compare it with Thouless energy since if they are not the in the ballistic case, this is the relevant parameter. E.g. plotting $I_c R_n$ vs. E_{th} could help, and where it shows a plateau could show where it enters ballistic or short regime.
- A small suggestion: it is really hard to between the symbols is in Fig 1g. Either the colours should be different, or less points should be used, or open symbols etc.
- If the authors think that there is a barrier between the proximity SC and the Al layer, they should indicate it in the paper. I don't think that without a barrier density of state effects would be enough to explain this.

Reply to Referee report for manuscript "Proximity effect, ideal superconducting diode and parity-conserving Cooper-pair transport in microwave-compatible Ge/SiGe heterostructures."

October 30, 2023

1 Reply to referee 1

We thank the referee for the careful analysis of our manuscript. We would like to point out that Fourier components of the SQUID, in our previous version, were extracted by using φ_1 , instead of the total phase drop across the SQUID. The latter is relevant only in case of high inductance and therefore does not lead to substantial differences. However, when considering the total phase drop φ the numerical simulation shows that the odd terms (b_1 and b_3) both vanish at half flux quantum and at the balanced regime also for finite conductance, as can be seen at the current version of Fig.ED9. We believe that this fact strengthens our manuscript even further. Therefore we have removed the sentence "However, the presence of a not negligible inductance might hinder the possibility of achieving parity conserving transport" from the resubmitted version of the paper.

2 Reply to referee 2

The authors have made effort to address all comments, but one critical point remains to be addressed. As the authors state, the novelty of the work is not in the materials development, but in the variation of the SiGe barrier thickness. The authors acknowledge that this reduction is a strong compromise in the uniformity of the quantum well, which is visible in for example a reduced mobility and percolation density. The sentence that the interface is crucial in determining the coherence seems to be at odds with the preceding section about dephasing times. However, I do agree with the authors that charge noise is most likely coming from the interface. Indeed the reduced uniformity and the reduced charge noise is the reason for the spin qubit community to focus on deeper quantum wells. I therefore remain of the option that the gain in the superconducting proximity effect is not sufficient enough compared to the reduction in uniformity to recommend this work for publication. Moreover, in my opinion this particular concern is not clearly addressed and a critical section highlighting all compromises and challenges is missing.

We thank the referee for their critical comment. We would like to emphasize that the focus of the present manuscript is on hybrid devices in planar Germanium and not on spin qubits. We remark that the gain in superconducting gap is one of the main results of our study, but not the only one. We highlight that we have presented a hybrid SQUID geometry, similar to arXiv:2304.00484 (now at Phys. Rev. Research 5, 033131 (2023)), demonstrating the presence of high-harmonics in the current-phase relation of Ge-based Josephson junctions. We furthermore tune the diode efficiency using microwaves, realizing the first experimental demonstration of an ideal diode ($\eta = 100\%$). Finally, we demonstrate of the possibility of realizing parity-conserving transport, simultaneously with arXiv:2306.05467, important for the implementation of parity-protected superconducting qubits.

Furthermore, we feel that from the first manuscript version we had pointed to the drawbacks of shallow quantum wells and possible mitigation strategies by including the following paragraph at the conclusions of our work:

"While the shallow QWs reported in this work are of limited mobility, possible mitigation strategies of this problem could include a careful engineering of the semiconductor/dielectric interface [1], including the use of

Ge caps [2], or growing the QWs on Ge instead of Si wafers [3].”

Following the comment of the referee we have slightly rephrased and extended this paragraph, which now reads:

”While the shallow QWs reported in this work are of limited mobility, which can be a challenge for the realization of scalable spin qubits, possible mitigation strategies of this problem could include a careful engineering of the semiconductor/dielectric interface [1], including the use of Ge caps [2], or growing the QWs on Ge instead of Si wafers [3]. A further solution could be to have a thin spacer in the areas where superconductivity should be induced and a thicker one in the areas where the spin qubits will be formed.”

Some minor points: In the outlook the sentence ’The reported large superconducting hard gap on a group IV material will pave the way towards spin qubit coupling via superconductor’ is not clear. A capacitive interaction can be sufficient for coupling to resonators. The authors mean hybrid systems based on tunnel coupling between superconductors and semiconductors.

We agree that a capacitive coupling is enough to realize two qubit gates with superconducting resonators. However, this sentence was referring to spin qubit coupling via superconductors as proposed by references 56 and 57 [4, 5], which have been cited in the mentioned by the referee sentence. In order to make this more clear we have rephrased the sentence which now reads ’The reported large superconducting hard gap on a group IV material will enable spin qubit coupling via superconductors’.

Figure R1: **Distribution of retrapping and switching currents.** Switching and retrapping currents have been recorded 100 times for a Josephson junction device with cap of 8 nm, $W = 2.4\mu\text{m}$ and $L = 120$ nm, at $V_g = -2\text{V}$.

3 Reply to referee 3

We thank the referee for their constructive criticism. Below we further clarify the open points.

In principle, the retrapping is also a stochastic process – one can measure retrapping current distribution. Do I understand well, that the authors claim, that the retrapping was reproducing well in their measurements?

Indeed, the retrapping current was reproducing very well in our measurements. This can be seen by comparing for example Fig. R2 in our previous reply, where the switching current was used, with Fig. 3b of the manuscript, for which the retrapping current was measured. Furthermore, we have analyzed another device in order to show the difference in the switching and retrapping current distribution (see Fig. R1).

I have suggested in the case of the $I_c R_n$ product discussion to compare it with Thouless energy since if they are not the in the ballistic case, this is the relevant parameter. E.g. plotting $I_c R_n$ vs. E_{th} could help, and where it shows a plateau could show where it enters ballistic or short regime.

We acknowledge that we forgot to comment on this point in our previous reply. We did not perform the analysis the reviewer had asked because of the following reasons. For calculating E_{th} one needs to know the mobility value. We have extracted it from Hall bar measurements for different values of carrier density/gate voltage. Similarly the $I_c R_n$ product has been extracted for different gate voltage values. However, the oxide which has been used on these two types of devices is different. For Hall bars which have dimensions of about $600\mu\text{m}$ a 20nm 300 degrees thermal aluminum oxide has been used in order to minimize leakage. On the other hand for the Josephson junction devices we have limited the thermal budget in order to avoid Al diffusion into the SiGe spacer. Therefore plasma assisted Aluminum Oxide of about 9-18nm has been deposited at 150 degrees. The consequence of this is that the same gate voltage value for Hall bars and Josephson junctions

does not correspond to the same carrier density. Therefore, we cannot correlate E_{th} and I_{cRn} .

A small suggestion: it is really hard to between the symbols is in Fig 1g. Either the colours should be different, or less points should be used, or open symbols etc.

Following the suggestion of the reviewer we have changed the colours of the traces.

If the authors think that there is a barrier between the proximity SC and the Al layer, they should indicate it in the paper. I don't think that without a barrier density of state effects would be enough to explain this.

Indeed we think that the SiGe spacer acts as a barrier through which superconductivity is induced from the Al into the Ge beneath it. We have added a sentence in the caption of Fig. 2, to point this out.

References

- [1] Degli Esposti, D. *et al.* Wafer-scale low-disorder 2deg in 28si/sige without an epitaxial si cap. *Applied Physics Letters* **120**, 184003 (2022).
- [2] Su, Y.-H., Chuang, Y., Liu, C.-Y., Li, J.-Y. & Lu, T.-M. Effects of surface tunneling of two-dimensional hole gases in undoped ge/gesi heterostructures. *Phys. Rev. Mater.* **1**, 044601 (2017).
- [3] Stehouwer, L. E. *et al.* Germanium wafers for strained quantum wells with low disorder. *arXiv preprint arXiv:2305.08971* (2023).
- [4] Leijnse, M. & Flensberg, K. Coupling spin qubits via superconductors. *Phys. Rev. Lett.* **111**, 060501 (2013).
- [5] Spethmann, M., Bosco, S., Hofmann, A., Klinovaja, J. & Loss, D. High-fidelity two-qubit gates of hybrid superconducting-semiconducting singlet-triplet qubits. *arXiv preprint arXiv:2304.05086* (2023).

REVIEWERS' COMMENTS

Reviewer #2 (Remarks to the Author):

In the previous round, I have made two comments. Regarding the first comment, the authors argue that their work is more than only a demonstration of an increased superconducting gap and I support this statement. I also agree with the authors that the focus is not on spin qubits. It is for that reason also important to highlight the compromises and challenges in a broader perspective. In particular because charge noise and decreased mobility are relevant to many experiments. In my opinion, the rephrased manuscript is not clearly reflecting this and I strongly encourage the authors to make a substantial change in this regard and write a more balanced manuscript.

Regarding the second comment, the authors write: 'The reported large superconducting hard gap on a group IV material will enable spin qubit coupling via superconductors'. Again, this is a statement that is already demonstrated using a capacitive interaction and a hard gap is not a requirement. I agree that the references motivate a different proposal, but that is not clear from this sentence. Please correct this.

Reviewer #3 (Remarks to the Author):

The authors have addressed all my remaining comments, therefore I suggest to accept the paper as it is.

Reply to Referee report for manuscript "Proximity effect, ideal superconducting diode and parity-conserving Cooper-pair transport in microwave-compatible Ge/SiGe heterostructures."

November 16, 2023

1 Reply to referee 2

In the previous round, I have made two comments. Regarding the first comment, the authors argue that their work is more than only a demonstration of an increased superconducting gap and I support this statement. I also agree with the authors that the focus is not on spin qubits. It is for that reason also important to highlight the compromises and challenges in a broader perspective. In particular because charge noise and decreased mobility are relevant to many experiments. In my opinion, the rephrased manuscript is not clearly reflecting this and I strongly encourage the authors to make a substantial change in this regard and write a more balanced manuscript.

Following the suggestion of the referee, we mention the issue of charge noise in the discussion of our manuscript.

Regarding the second comment, the authors write: 'The reported large superconducting hard gap on a group IV material will enable spin qubit coupling via superconductors'. Again, this is a statement that is already demonstrated using a capacitive interaction and a hard gap is not a requirement. I agree that the references motivate a different proposal, but that is not clear from this sentence. Please correct this.

We rephrased it to: "The reported large superconducting hard gap on a group IV material will enable spin qubit coupling via coherent tunneling and cotunneling processes that involve (crossed) Andreev reflection."

We thank all the referees for their careful analysis of our manuscript.

References